



# A Combined Gas- and Particle-phase Analysis of Highly Oxygenated Organic Molecules (HOM) from α-pinene Ozonolysis

Jian Zhao[1], Ella Häkkinen[1], Frans Graeffe[1], Jordan E. Krechmer[2,a], Manjula R. Canagaratna[2], Douglas R. Worsnop[2], Juha Kangasluoma[1], and Mikael Ehn[1]

[1]Institute for Atmospheric and Earth System Research/Physics, Faculty of Science, University of Helsinki, Helsinki, 00014, Finland.
[2]Aerodyne Research Inc., Billerica, Massachusetts, 01821, United States.
[a]Now at: Bruker Daltonics Inc., Billerica, Massachusetts, 01821, United States

*Correspondence to*: Jian Zhao (jian.zhao@helsinki.fi) and Mikael Ehn (mikael.ehn@helsinki.fi)

**Abstract.** Highly oxygenated organic molecules (HOM) are important for the formation of secondary organic aerosol (SOA), which poses serious health risks and exerts great influence on Earth's climate. However, the speciation of particle-phase HOM and its relationship with gas-phase HOM formation has been limited by the lack of suitable analytical techniques. Here, combining a novel particle evaporation inlet VIA (Volatilization Inlet for Aerosols) with a nitrate chemical ionization mass spectrometer ($NO_3$-CIMS), gas- and particle-phase HOM products of α-pinene ozonolysis were studied under different conditions. Within the 50-min residence time of our Teflon chamber, we observed enhancement of $C_{16}$-$C_{19}$ HOM dimers in particles compared to the HOM that were condensing. In particular, gas-phase dimer formation was considerably suppressed in experiments with the addition of CO or NO, but dimers still made up a considerable fraction of the observed SOA. In addition to the generally shorter carbon skeletons of the particle phase dimers (i.e. $C_{16}$-$C_{19}$) compared to the gas phase ($C_{19}$-$C_{20}$), average O/C ratios of the HOM (especially in the dimer range) also decreased slightly in the particle phase. $C_{17}H_{26}O_z$ compounds, which have often been reported by previous offline measurements, dominate the particle-phase HOM mass spectra in α-pinene ozonolysis experiments. Our results indicate that these $C_{17}$ compounds might be related to particle-phase processes within one hour after HOM condensation. However, the new VIA-$NO_3$-CIMS system used in this work will require more detailed characterization to better understand how the thermal desorption and wall effects may modify the measured particle-phase HOM distributions. Nevertheless, for example organic nitrate measured by this novel VIA-$NO_3$-CIMS system was consistent with the measurements of an Aerodyne Aerosol Mass Spectrometer (AMS), showing the capability of this system as a promising technique for particle-phase HOM measurements. Taken together, we believe that this system is a promising technique for combined online gas- and particle-phase HOM measurements.

## 1 Introduction

Secondary organic aerosol (SOA), formed from condensable oxidation products of volatile organic compounds (VOCs), contributes a large fraction to tropospheric fine particles, which greatly influence human health and global climate (Mauderly and Chow, 2008; Jimenez et al., 2009; Boucher et al., 2013). Recently, highly oxygenated organic molecules (HOM) formed through rapid autoxidation (i.e. consecutive intramolecular H-shifts followed by $O_2$ addition) of peroxy radicals ($RO_2$) were found to play an important role in the growth and/or formation of new particles in the atmosphere (Crounse et al., 2013; Ehn et al., 2014; Kulmala et al., 2014; Jokinen et al., 2014; Kirkby et al., 2016; Bianchi et al., 2019). $RO_2$ are ubiquitous intermediates formed from the attack of various oxidants on VOC precursors, and therefore the fate of $RO_2$ and related formation pathways toward HOM is crucial to understanding the gas-phase chemistry and corresponding SOA formation in Earth's lower atmosphere (Orlando and Tyndall, 2012; Ehn et al., 2017).

Monoterpenes ($C_{10}H_{16}$), emitted from various terrestrial vegetation systems, account for ~15% of biogenic VOCs emissions (by mass) at a global scale (Guenther et al., 2012), and may contribute largely to fine organic aerosols (Ding et al., 2014; Zhang et al., 2018; Mcfiggans et al., 2019). α-pinene (AP) is one of the most important monoterpenes, which has been shown to form $C_{10}$ monomers and $C_{20}$ dimers as HOM products (Ehn et al., 2014). An endocyclic double bond and relatively large carbon skeleton increases the yield of these low volatile HOM products compared to many other biogenic VOCs (e.g.



isoprene $C_5H_8$, the largest biogenic gas precursor), making it important in terms of SOA formation (Donahue et al., 2012; Donahue et al., 2013; Jokinen et al., 2015; Kurten et al., 2016). Furthermore, the ozonolysis of α-pinene produces aldehydic

H atoms, which greatly promotes H-shifts in the $RO_2$, and consequently increases HOM yields (Mentel et al., 2015; Rissanen et al., 2014; Bianchi et al., 2019). Consistently, HOM yields from other oxidants (e.g. OH, $NO_3$, and Cl) reacting with α-pinene are typically lower (Jokinen et al., 2015; Berndt et al., 2016; Kurtén et al., 2017; Wang et al., 2020). These recent HOM yield findings are also consistent with earlier SOA mass yields, which have showed that $O_3$ forms more SOA than the other oxidants (Hoffmann et al., 1997; Griffin et al., 1999; Presto et al., 2005).

The understanding of HOM formation from ozonolysis of α-pinene has gradually improved during the past decade, owing to the developments/improvements of state-of-the-art mass spectrometers and quantum chemical calculations (Ehn et al., 2012; Vereecken et al., 2015; Riva et al., 2019; Bianchi et al., 2019; Iyer et al., 2021). The nitrate chemical ionization inlet combined with an atmospheric pressure interface time-of-flight mass spectrometer ($NO_3$-CIMS for short hereafter) has been widely used to detect gas phase HOM as adduct clusters with $NO_3^-$ with high sensitivity and selectivity (Junninen et al.,

2010; Jokinen et al., 2012; Ehn et al., 2014; Hyttinen et al., 2017). Other reagent ions (e.g. $CH_3COO^-$, $C_3H_7NH_3^+$, and $NH_4^+$) have also been used with different sensitivities/selectivities to increase our understanding of HOM species and autoxidation (Berndt et al., 2016; Berndt et al., 2018; Riva et al., 2019). However, the characterization of HOM in the particle phase as well as the physicochemical links between gas- and particle-phase HOM are still not clear. One of the difficulties is the lack of suitable analytical techniques to identify the labile peroxides in α-pinene SOA (Docherty et al., 2005; Reinnig et al., 2009;

Krapf et al., 2016), which are formed literally after each H-shift of $RO_2$ during HOM formation in the gas phase (Crounse et al., 2013; Mentel et al., 2015; Wang et al., 2017). The lifetime of organic peroxides in α-pinene SOA was reported to be around one hour through dark decomposition (Krapf et al., 2016). Formation of high molecular weight ester dimers with aldehydes/ketones through the Baeyer-Villiger reactions was proposed as another particle-phase loss of these hydroperoxides (Claflin et al., 2018). The concomitant formation of carboxylic acids through the Baeyer-Villiger reactions is consistent with

the fact that a large portion of organic acids was observed in ambient SOA (Hallquist et al., 2009; Yatavelli et al., 2015). However, the experiments conducted by Claflin et al. (2018) were under conditions of high VOC concentrations (corresponding 3 mg m$^{-3}$ SOA), and whether the proposed reactions can happen under ambient conditions is still unknown. Another study suggested that some HOM species (assuming being keto-hydroperoxides) may undergo decomposition through the Korcek mechanism in the particle phase, which forms $C_7$-$C_9$ carboxylic acids and $C_1$-$C_4$ carbonyls (Mutzel et

al., 2015).

Here, a novel particle evaporation inlet named VIA (Volatilization Inlet for Aerosols, Aerodyne Research, Inc.) combined with a $NO_3$-CIMS was used to detect HOM in the particle phase at trace concentrations. There are several special features of this system compared to currently used/developed techniques (Häkkinen et al., 2022). Traditional molecular level SOA measurements were usually conducted in an offline or semi-online way, during which filter/metal surfaces are necessary to

collect particles (Hallquist et al., 2009; Yatavelli et al., 2012; Mutzel et al., 2015). We tried to minimize unwanted surface reactions of SOA by heating it directly within the sampling flow after removing the gases, which is important for the detection of labile organic compounds. Another online technique that was recently develop is the Extractive Electrospray Ionization (EESI) inlet combined with a mass spectrometer, aim at providing rapid online molecular detection of SOA (Lopez-Hilfiker et al., 2019; Pospisilova et al., 2020) and updated to a dual-phase inlet by Lee et al. (2022). The EESI has

proven very useful for SOA detection, but it cannot provide comparable gas-phase measurements. Both gas- and particle-phase organic molecules could be detected by a Filter Inlet for Gases and Aerosols (FIGAERO) coupled with an iodide/acetate CIMS (Lopez-Hilfiker et al., 2014; Lopez-Hilfiker et al., 2015) or a Chemical Analysis of Aerosol Online (CHARON) inlet coupled with a Proton-Transfer-Reaction (PTR) mass spectrometer (Eichler et al., 2015; Müller et al.,





2017). These are robust and powerful systems for online (CHARON) or semi-online (FIGAERO) SOA measurements,
though each reagent ion and setup provides its own selectivity and sensitivity, with neither being specifically selective
towards HOM (Riva et al., 2019). Recently, a Thermal Desorption-Differential Mobility Analyzer (TD-DMA) coupled with
a $NO_3$-CIMS was deployed to detect nano (~10-30 nm) SOA particles in a semi-online approach (Wagner et al., 2018;
Caudillo et al., 2021). Similar to our system, nitrate was used as the reagent ion for both gas- and particle-phase
measurements, reducing ambiguities when comparing HOM species measured in the gas versus particle phase. The
advantage of the TD-DMA system is that it uses a DMA to size-selected particles, but the drawback is the lowered (bulk)
sensitivity due to the need for charging in the DMA. Thus, the novel VIA inlet coupled with a $NO_3$-CIMS will be a promising
system to identify HOM species in the particle phase, using a fully online approach.

In this work, in order to provide molecular speciation of HOM in the particle phase and deduce its dependence on HOM
formed in the gas phase, we systematically studied HOM products in both phases using the novel VIA-$NO_3$-CIMS system.
From our earlier experiments, higher dimer than monomer concentrations were observed in the particle phase for four
different VOCs using this system (Häkkinen et al., 2022), in contrast to the measurements in the gas phase. However, those
experiments were mainly focused on the particle phase and were conducted with high precursor conditions using a Potential
Aerosol Mass (PAM) reactor (Kang et al., 2007; Lambe et al., 2011). Here, starting with relatively lower concentrations of
α-pinene and $O_3$ (dark reactions) as our base case in a Teflon chamber, CO or $NO_2$/NO was introduced into this reaction
system to change the distribution of HOM products, meanwhile investigate the impacts in both the gas and particle phases.
HOM products were grouped according to different carbon numbers, along with dimer to monomer ratios (D/M), to compare
their distribution in different phases. In addition, bulk chemical compositions (O/C, H/C, and N/C) of HOM in the particle
phase were estimated and compared to a widely used Aerodyne Aerosol Mass Spectrometer (Decarlo et al., 2006).

## 2 Experimental section

### 2.1 Experimental setup

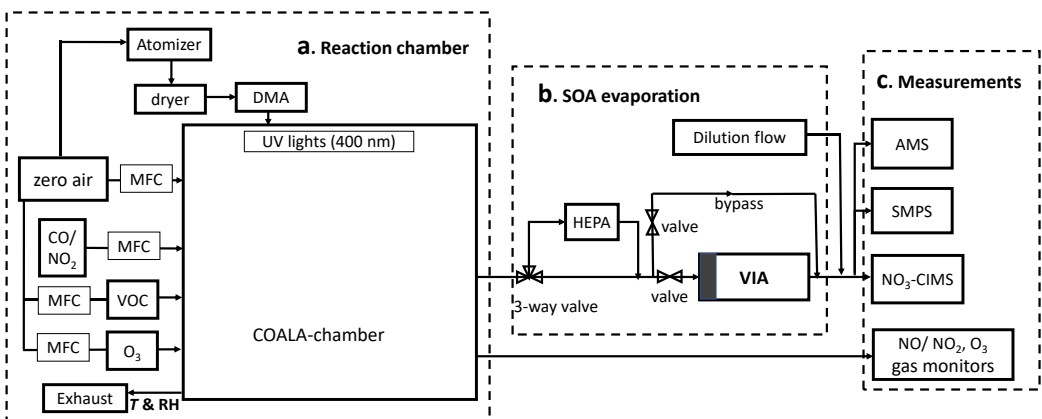

**Figure 1. Setup for the thermal evaporation experiments used in this work. Note that all three parts (a, b, and c) are necessary for the particle-phase HOM measurements, whereas only part a and c are used for the gas-phase HOM measurements (i.e. all instruments sampling directly from the chamber).**

The setup for experiments conducted in this work includes three parts as shown in Figure 1. a) Reaction chamber (2 $m^3$,
Teflon), in which HOM and SOA were formed from α-pinene ozonolysis. The experiments were operated in a continuous





flow mode (residence time ~50 min) under dry conditions (RH < 2 %) and at room temperature (26 ± 1 ℃). The total inflow of 40-L min⁻¹ cleaned air was purified by a clean air generator (AADCO model 737-14, Ohio, USA). α-pinene was injected using a syringe (Hamilton) and a syringe pump (Cole Pamer, IL, USA). $O_3$ was generated by an ozone generator (Dasibi

1008-PC). Sodium chloride solution (NaCl in Milli-Q water) was used to generate seed particles (80 nm) to make the HOM condense onto larger particles rather than forming new (small) particles. NaCl was used instead of the more common ammonium sulfate (AS) to prevent the potential depleting of reagent ions by large amounts of sulfuric acid, and to avoid its nucleation with evaporated HOM vapors. CO or $NO_2$ was injected into the chamber from a gas cylinder during some of the experiments to change the gas-phase HOM chemistry. NO could be generated from photolysis of $NO_2$ with UV lights (400

nm, LEDLightmake.inc). The entire flow system was controlled by a series of mass flow controllers (MKS, G-Series, 0.05-50 L min⁻¹, Andover, MA, USA). For more detailed descriptions of the "COALA" chamber system and facilities at the University of Helsinki, please refer to Riva et al. (2019) and Peräkylä et al. (2020).

b) SOA evaporation system, a novel VIA inlet was deployed to evaporate particles for analysis. First, the gaseous vapors were removed from the sample flow (1.5-2 L min⁻¹) by a honeycomb-activated charcoal denuder (shaded region in Fig.1,

part b) in front of the heating region within the VIA. The remaining particles were then heated to a chosen temperature (25 to 300 ℃, measured from the surface of the heating tube by a thermocouple) causing evaporation of molecules in the particles. After the VIA, a dilution flow (10 L min⁻¹) was introduced, in part to cool down the sample before entering the $NO_3$-CIMS, and to provide enough flow for all instruments.

Based on our tests, the removal efficiency of this charcoal denuder depends on the flow rate through VIA as well as the

concentrations and types of the target gas. The best removal performance was obtained with a small flow rate (1-2 L min⁻¹). We observed that >95% of ~4 ppm $O_3$ and >80% of 500-800 ppb α-pinene were removed at 1.5 L min⁻¹. Thus, the gas denuder worked efficiently under low concentrations used in this work (Table 1) and gas-phase reactions within the VIA were assumed to be negligible. The particle transmission efficiency of VIA was ~90% tested with room air (10-200 nm particles). The denuder was replaced with a new/cleaned one before the start of each experiment to prevent potential

contaminations, and it can be regenerated by heating to 100 ℃ in a clean air flow for about four hours. Using valves, we could either pass the sampled air through a high efficiency particulate air (HEPA) filter to remove particles and through the VIA to remove the gaseous vapors for a background measurement, or bypass the HEPA and VIA completely to measure the unperturbed sample. However, to avoid the additional losses and dilution, direct gas-phase HOM measurements were done by completely bypassing part b) of the setup. A more detailed characterization of this VIA-$NO_3$-CIMS system was

investigated in a previous work (Häkkinen et al., 2022).

c) Measurements, a suite of gas and particle phase instruments were deployed. Gas phase and evaporated HOM from SOA were measured by a $NO_3$-CIMS (Tofwerk AG/Aerodyne Research, Inc.), which is highly sensitive to detect HOM as adduct clusters with $NO_3^-$ (Junninen et al., 2010; Jokinen et al., 2012). Nitrate ions were generated from nitric acid ($HNO_3$) through an X-ray source. The concentrations of HOM were estimated by applying a calibration factor $C_x$ (= $2 \times 10^{10}$ cm⁻³) to the raw

signals after being normalized by the sum of reagent ions (i.e. nitrate monomer, dimer, and trimer). This calibration factor was estimated using the sulfuric acid signals obtained by the VIA-$NO_3$-CIMS system from several nanograms of AS assuming 70% evaporation at 300℃ (Häkkinen et al., 2022). Thus, a large uncertainty comes with this estimation, along with other factors including the total losses of HOM during evaporation and sampling. We expect the uncertainty be at least a factor 3 (+200%/-67%), and therefore the absolute concentrations are only reported in this work to give rough

approximations of the sampled concentrations. SOA mass was measured by a long time-of-flight Aerosol Mass Spectrometer (LTOF-AMS, Aerodyne Research, Inc.), using thermal evaporation (600 ℃) and electron impact ionization (70 eV) techniques (Decarlo et al., 2006; Canagaratna et al., 2007). The (relative) ionization efficiency of AMS towards inorganic



species was calibrated with 300 nm ammonium nitrate (AN) and AS particles (Jayne et al., 2000). The particle size distribution (10-500 nm) was measured with a scanning mobility particle sizer system (SMPS, consisting of one 3081 long

differential mobility analyzer and one 3750 condensation particle counter, TSI). $O_3$, NO, and $NO_2$ were monitored by a UV photometric analyzer (Model 49P, Thermo Environmental Instruments), an Eco Physics gas analyzer (Model CLD 780 TR), and a chemiluminescence NO–$NO_2$–$NO_x$ analyzer (Model 42i, Thermo Fisher Scientific), respectively. In addition, a proton transfer reaction time-of-flight mass spectrometer (PTR-TOF 8000, Ionicon Analytik Gmbh) was used to measure α-pinene. Unfortunately, it worked intermittently and was only connected to the chamber twice during the first experiments. The

concentrations of α-pinene in the first experiment were evaluated based on 5-min averaged signals from the data acquisition panel applying the sensitivity obtained from the calibration conducted that morning (two brown cross markers in Fig. 2b). For the rest experiments, we used the syringe pump injection rates to estimate the injected concentrations of α-pinene.

**2.2 Chamber experiments**

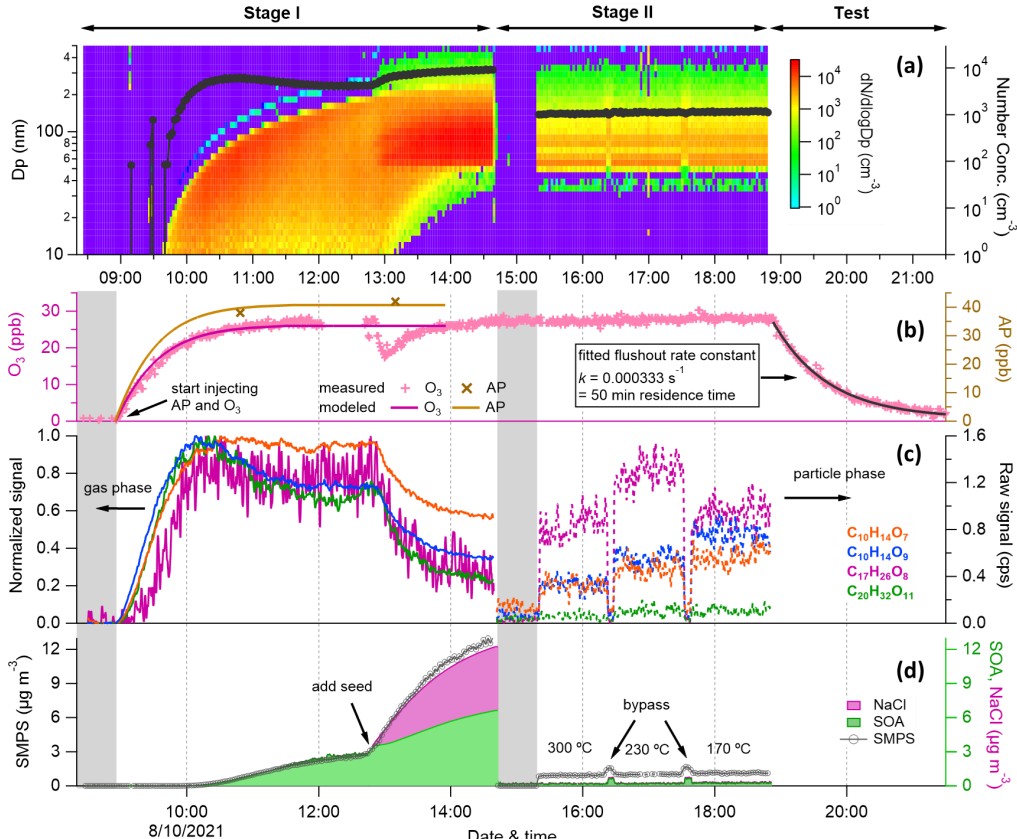

**Figure 2. Overview of experiment No. 1 (input flow with 53 ppb α-pinene and 33 ppb $O_3$). (a) Particle number concentration and size distribution sampled by SMPS as a function of time. Time series of (b) measured and modeled ozone and α-pinene concentration (1-min averaged) in the chamber, (c) gas-phase (solid lines, normalized to their maximums) and particle-phase (dashed lines, raw signals) HOM species (10-s averaged), (d) total aerosol, organics, and sodium chloride mass concentrations from SMPS (2.2-min averaged, black circles) and AMS measurements (20 s averaged). The first and second shaded areas are gas**

**phase and particle phase background measurements, respectively. Note that the generally lower particle phase measurements in stage II compared to stage I is owing to the addition of a dilution flow (Fig. 1), and no measurements given above were corrected for chamber wall loss.**





A typical experiment was conducted as shown in Figure 2, including two stages. In stage I, all instruments sampled directly from the chamber (i.e. part b in Fig. 1was removed from the system). The COALA chamber was flushed with purified air

for at least 12 hours before each experiment (chamber background was determined during this period, first shaded area in Fig. 2). HOM formed rapidly once α-pinene and $O_3$ were injected into the chamber. SOA particles took ~40 minutes to form and grow larger than 10 nm to be measured by an SMPS, whereas it took about 1.5 hours for the AMS to capture the SOA formation. Then, NaCl seed particles were added into the chamber after HOM and SOA concentrations stabilized, which can be seen as a dramatic increase of the particle mass in Figure 2d. However, the thermal desorption temperature of 600 ºC

used by the AMS was not enough for effective NaCl evaporation/measurements, resulting in discrepancies in aerosol mass concentration compared to the SMPS (Fig. S1). The mass concentrations shown in Fig. 2d were modified based on AMS and SMPS measurements and the method to reconstruct SOA and NaCl mass is explained in the Supplementary (section S1). In contrast to the aerosol mass, gas-phase HOM decreased upon seed addition owing to enhanced condensation onto the newly added particles that increased the condensation sink. By calculating the difference in gas-phase HOM spectra

before and after seed particle addition, we can deduce the condensing HOM products. We will use the term "condensed gas phase" hereafter to directly compare the condensing HOM to the particle-phase HOM measurements. In stage II, the VIA was mounted manually between the chamber and instruments for particle-phase HOM measurements as shown in Figure 1. After background measurements for the particle phase (second shaded area in Fig. 2), three different temperatures (170, 230, and 300 ºC) were set for the VIA heater, each one kept for about one hour for data collection. Note that the analysis and

following discussions mainly focused on the results of 230 ºC, which was found to give the highest sensitivity/signals for total particle-phase HOM measurements with less losses or thermal decomposition compared to other temperatures. However, as shown in Fig. 2c, larger molecules ($C_{17}$ and $C_{20}$ compounds) were found to evaporate at higher temperatures, while signals of smaller or more volatile molecules ($C_{10}$ compounds) started to decrease at the highest temperatures. This effect is consistent with the previous results reported by Häkkinen et al. (2022) and needs to be kept in mind when

interpreting the particle-phase HOM mass spectra. Between the change of temperatures, the sampling line bypassed VIA to obtain SOA mass concentration without heating. Then, the evaporated SOA mass could be estimated by taking the difference of measurements between bypass and VIA modes. At the end of the experiment, a test stage was included for determining kinetic parameters for use in a box model (described in section 2.4), such as the flush-out rate (as shown in Fig. 2b) and the photolysis rate of $NO_2$. All experiments conducted in this work are summarized in Table 1 and will be explained in section

200 2.4.

**Table 1. Experimental Conditions for the COALA Chamber Experiments. The steady state conditions refer to the stabilized concentrations before adding seed particles.**

| No. | Input flow (in) conditions | | | | | steady state (ss) conditions | | | | |
| | $[AP]_{in}$ (ppb) | $[O_3]_{in}$ (ppb) | $[NO_2]_{in}$ (ppb) | [CO] (ppm) | UVlights (400 nm) | $[AP]_{ss}$ (ppb) | $[O_3]_{ss}$ (ppb) | $[NO_2]_{ss}$ (ppb) | $[NO]_{ss}$ (ppb) | $[SOA]_{ss}$ (µg m$^{-3}$) |
|---|---|---|---|---|---|---|---|---|---|---|
| 1 | 53 | 33 | 0 | 0 | off | 41 | 26.5 | 0 | 0 | 2.84 |
| 2 | 80 | 33 | 0 | 0 | off | 63.6 | 24.0 | 0 | 0 | 7.01 |
| 3 | 120 | 33 | 0 | 0 | off | 98.3 | 20.5 | 0 | 0 | 11.83 |
| 4 | 120 | 33 | 0 | 100 | off | 101.2 | 20.2 | 0 | 0 | 9.30 |
| 5 | 120 | 33 | 7.6 | 0 | off | 97 | 21.9 | 5.3 | 0 | 8.40 |
| 6 | 120 | 33 | 7.7 | 0 | on | 96.2 | 33.1 | 3.3 | 0 | 10.9 |
| 7 | 120 | 33 | 17.3 | 0 | on | 95.3 | 46.6 | 8.1 | 0 | 13.95 |
| 8 | 120 | 33 | 46.6 | 0 | on | 92.9 | 66.0 | 28.5 | 1.4 | 7.50 |
| 9 | 53 | 33 | 46.3 | 0 | on | 38.9 | 51.6 | 35.4 | 2.5 | 1.34 |



### 2.3 Data analysis

The mass spectrometry data sets were analyzed using Igor Pro (WaveMetrics, Inc., USA) based packages (Tofware_3_2_0 and ToF_AMS_HRAnalysis_v1_25A). Based on the mechanisms of HOM formation, "HOM" was suggested by Bianchi et al. (2019) to describe compounds directly related to gas phase autoxidation (Crounse et al., 2013) in atmospheric conditions and with at least six oxygen atoms, so that could be effectively detected by a $NO_3$-CIMS as used in this work. Although organic compounds with O numbers less than six (e.g. norpinic acid $C_8H_{12}O_4$ and pinic acid $C_9H_{14}O_4$) were largely observed in both the gas and particle phase in α-pinene ozonolysis (Lopez-Hilfiker et al., 2015; Zhang et al., 2015), these compounds would not be regarded as HOM, nor would they be effectively detected by a $NO_3$-CIMS (Hyttinen et al., 2015; Hyttinen et al., 2017). For experiments with $NO_x$, measured HOM signals were separated into two groups as $HOM_{CHO}$ ($C_xH_yO_z$ compounds) and N-containing $HOM_{ON}$ ($C_xH_yN_{1-2}O_z$ compounds), based on the number of N atom in the fitted molecular formula excluding the reagent ion ($NO_3^-$). The peak positions located at odd or even nominal mass also helped to separate these compounds. Note that the contribution of N-containing peaks was only considered for $NO_x$ experiments, owing to the relatively low contribution of nitrate dimer ($HNO_3 \cdot NO_3^-$) charged signals to the total signals (4.3±1.5%) for α-pinene ozonolysis experiments. In addition, HOM products were grouped according to different carbon numbers to investigate their distribution. $C_8$-$C_{10}$ and $C_{16}$-$C_{20}$ compounds were regarded as HOM monomer and dimer, respectively, to be consistent with previous research works (Zhao et al., 2018b; Molteni et al., 2019). Thus, HOM species discussed in this study were limited to molecules with a formula as $C_{x>7}H_{y>7}N_{0-2}O_{z>5}$ (excluding the $NO_3^-$ reagent ions), which contributed 83-91% of the total measured signals (shaded areas in Fig. 3).

Finally, elemental ratios were calculated in order to compare the bulk chemical composition of SOA measured by the AMS and the VIA-$NO_3$-CIMS system. Molar O/C, H/C, and N/C ratios of AMS were calculated using the Improved-Aiken method (IA), with uncertainties of 28%, 13%, and 22%, respectively (Aiken et al., 2008; Canagaratna et al., 2015). Nitrate signals measured by the AMS are almost all from organic nitrates, which are consistent with the estimation of $NO_x^+$ ratio method (Farmer et al., 2010; Day et al., 2022). Molar O/C, H/C, and N/C ratios of the $NO_3$-CIMS measurements (signal weighted) were calculated based on the fitted chemical formulas. The uncertainty mainly comes from the accuracy of high-resolution fitting results, relative calibration factors among different HOM compounds (instead of the absolute calibration factors), and the signal-to-noise ratios of each compound.

### 2.4 Kinetic model and $RO_2$ chemistry

A simple 0-D kinetic box model was used to model α-pinene concentrations and support the interpretation of the experimental results. The chemical reactions and physical parameters are described in Section S2. As shown in Figure 2b, the modeled α-pinene is consistent with the PTRTOF measurements. Based on the only two α-pinene measurements during the first experiment, modeled α-pinene concentrations were estimated with an accuracy of 10%, though this will probably be less accurate for the $NO_x$ experiments because more reactions need to be considered in the model. The main reactions that will be discussed in the following sections are listed below:

$RO_2 \rightarrow RO_4 \rightarrow RO_6 \rightarrow \cdots$ (R1a, autoxidation)

$RO_2$ (H-shift) $\rightarrow R(=O)H + OH$ (R1b, unimolecular termination)

$RO_2 + RO_2 \rightarrow R\text{-}OH + R(=O)H + O_2$ (R2a, carbonyl and hydroxyl)

$RO_2 + RO_2 \rightarrow RO + RO + O_2$ (R2b, alkoxy radical)

$RO_2 + RO_2 \rightarrow ROOR + O_2$ (R2c, dimer)

$RO_2 + HO_2 \rightarrow ROOH + O_2$ (R3a, hydroperoxide)





$$RO_2 + HO_2 \rightarrow RO + OH + O_2 \qquad \text{(R3b, alkoxy radical and OH)}$$

$$RO_2 + NO \rightarrow RONO_2 \qquad \text{(R4a, organic nitrates)}$$

$$RO_2 + NO \rightarrow RO + NO_2 \qquad \text{(R4b, alkoxy radical and NO}_2\text{)}$$

Peroxy radicals are the key intermediate in α-pinene ozonolysis leading to HOM formation (Ehn et al., 2017; Iyer et al., 2018; Berndt et al., 2018). They can undergo unimolecular autoxidation forming more oxidized $RO_2$ (Eq. R1a) or termination forming monomer products (Eq. R1b). They can also undergo bimolecular reactions (Eq. R2,3,4) with $RO_2/HO_2/NO$ to form closed-shell monomer/dimer products or alkoxy radicals (RO), which may or may not form another $RO_2$ to restart this cycle (Mentel et al., 2015; Berndt et al., 2018; Bianchi et al., 2019). The branching ratios depend on the concentration of reaction partners.

In α-pinene ozonolysis, the fate of $RO_2$ is mainly through unimolecular (Eq. R1a,b) and bimolecular (Eq. R2a,b,c) reactions. Different α-pinene concentrations will change the relative importance of these two kinds of pathway. Thus, the concentration-dependent distribution of HOM products in both phases were investigated in experiment No. 1-3 (α-pinene, 53-120 ppb). In experiment No. 4 with CO as an OH scavenger, ozonolysis was largely isolated so that we could focus on

$O_3$-derived $RO_2$ and corresponding HOM products. Note that added CO will turn OH to $HO_2$, thus making $HO_2$ a competitive partner to react with $RO_2$ (Eq. R3a,b). In experiment No. 5, $NO_2$ was added under dark conditions to introduce $NO_3$ (formed from the reaction of $NO_2$ and $O_3$) initiated oxidation forming nitrooxy peroxy radicals ($nRO_2$), which has been proposed to not undergo autoxidation very efficiently (Kurtén et al., 2017). Nevertheless, the fate of $nRO_2$ through bimolecular (Eq. R2a,b,c) reactions with another $RO_2/nRO_2$ would still be expected. Finally, NO was formed after turning on the UV lights

(experiment NO. 6-9, inflow VOC/$NO_x$ = 15.6-1.1), which partly converted $RO_2$ to organic nitrates as N-containing HOM monomers (Eq. R4a) instead of through bimolecular (Eq. R2a,b,c) reactions with another $RO_2$. Relatively lower $O_3$ to α-pinene ratios (0.28-0.62) than their distribution in the real atmosphere were used in this work to reduce reactions between NO and $O_3$, which maximizes bimolecular reactions between NO and $RO_2$. Note that the gas-phase HOM were measured in experiments No. 6-9 with both lights off and on, whereas the particle phase measurements were only obtained with the lights

on. Experiment No.5 was conducted with lights off during the entire experiment, which is the only one containing the information about particle-phase HOM products from $NO_3$ and α-pinene reactions without NO chemistry. In the following sections, both the identification of gas and particle phase HOM products and their relationship will be investigated in detail for each reaction system.

## 3. Results

### 3.1 HOM distribution under different conditions

### 3.1.1 α-pinene and $O_3$ dark reactions

Mass defect plots of particle-phase signals measured by the VIA-NO$_3$-CIMS system from α-pinene ozonolysis are given in Fig. 3 a-c, which showed large differences compared to gas phase measurements (Fig. S2). In the gas phase, $C_{10}H_{14,16}O_z$ dominates HOM monomers followed by $C_8H_{12}O_z$, while the dimers (much lower than monomers) were almost evenly spread

among $C_{17}$-$C_{20}$ compounds (Fig. 3d). However, in the particle phase, dimers (dominated by $C_{17}H_{26}O_z$ followed by $C_{18,19}H_{26,28}O_z$) showed comparable or even higher contribution to total HOM signals than monomers (mainly $C_{10}$ compounds). Similarly, higher dimer than monomer signals of α-pinene SOA was measured in sodiated ion adducted ([M + Na]+, electrospray ionization) mass spectra (Zhang et al., 2017). In addition, $C_{16}H_{24,26}O_z$ was almost exclusively observed in the particle phase, indicating particle-phase related sources of these compounds. Oxygen numbers also changed, peaking




around $O_6$-$O_7$ in the gas phase but shifted to $O_7$-$O_{10}$ in the particle phase (Fig. 3f). Nevertheless, the signal-weighted O/C of

the particle-phase HOM is lower than that of the gas phase (details in section 3.2), owing to large shift of C numbers in the

particle phase. Last, H numbers (Fig. 3d) are highly correlated with C numbers for HOM products both in the gas and particle

phases, as expected. And a small amount of $C_{10}H_{15}O_z$ and $C_9H_{13}O_z$ peroxy radicals were measured in the gas phase but were

negligible in the particle phase.

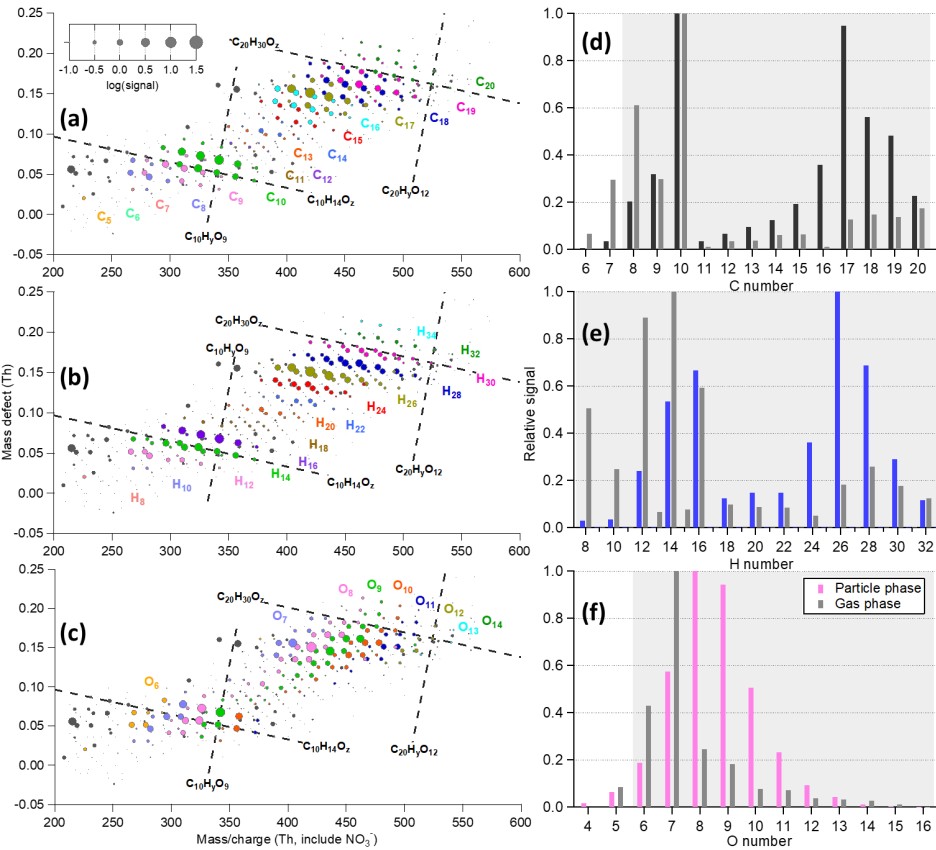


**Figure 3. (left) Mass defect plot of particle phase signals (60-min average, 230 ºC) measured in experiment No. 3 (input flow concentration as 120 ppb α-pinene and 33 ppb $O_3$). The data points are identical in panels a-c, but colored by the (a) carbon number, (b) hydrogen number, and (c) oxygen number. The area of the markers is proportional to the logarithm of measured raw signals. The grey markers are either unknown peaks (unidentified) or not HOM compounds according to the nomenclature of**
**"HOM" (Bianchi et al., 2019). (right) Distribution of signals (normalized to the largest peak) grouped by (d) carbon number, (e) hydrogen number, and (f) oxygen number. The particle phase measurements are shown as colored bars, while the gas phase measurements are shown in grey. The shaded areas in panels d-f are the focus of this study.**

Many elemental compositions identical to the monomers and dimers identified in this work were also reported by other mass

spectrometry techniques (summarized in Table S2), though in general, species reported in this work contained more oxygen

than in previous publications (Müller et al., 2009; Zhang et al., 2015; Mutzel et al., 2015; Kristensen et al., 2020; Pospisilova

et al., 2020). For example, ester dimers with molecular formulas as $C_{17}H_{26}O_7$ and $C_{17}H_{26}O_8$ were identified by an Ultra-High

Performance Liquid Chromatography/Electrospray Ionization quadrupole Time-of Flight Mass Spectrometer (UHPLC/ESI-

qToF-MS) (Kristensen et al., 2020; Kahnt et al., 2018). However, the structure of many ester dimers with 5-7 oxygen atoms

were proposed to contain carboxylic functional groups at both ends (Kristensen et al., 2016; Zhang et al., 2015), which would



possibly be "invisible" by a $NO_3$-CIMS owing to large difficulties to form two hydrogen bonds with the reagent ions
       (Hyttinen et al., 2015). It remains unclear to what extent the identical elemental compositions reflect identical isomeric
       structures, given the differences between the methods. Nevertheless, the similarities in detected compositions are
       encouraging, even though one needs to be careful to not draw direct links to previous observations using different methods.

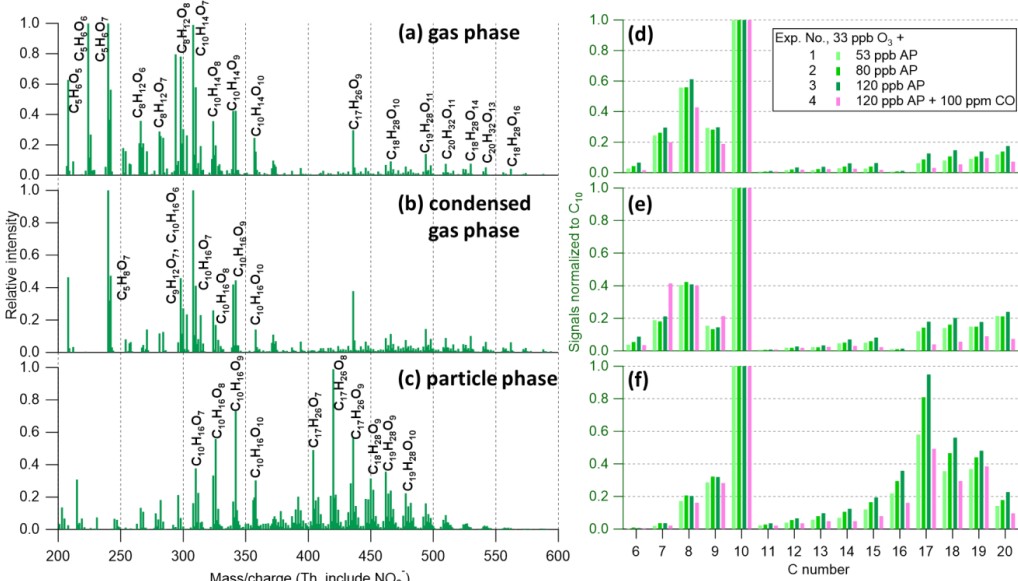

**Figure 4. (left) Mass spectra of (a) gas phase (without seed particles), (b) condensed gas phase (from the difference between gas-phase mass spectra without and with seed particles, see section 2.2), and (c) particle phase (at 230 ºC) measurements from**
**experiment No.3 (input flow concentration as 120 ppb α-pinene and 33 ppb $O_3$). Gas phase and condensed gas phase mass spectra**
       **(30-min average) are normalized to $(C_{10}H_{14}O_7)NO_3$ at m/z 308 with m/z 224 and m/z 240 off the scale. Particle phase mass spectra**
       **(60-min average) is normalized to $(C_{17}H_{26}O_8)NO_3$ at m/z 420. (right) The distribution of HOM products for experiments No. 1-4,**
**grouped by C numbers, are summarized in (d) for gas phase, (e) for condensed gas phase, and (f) for particle phase. Signals are**
       **normalized to the $C_{10}$ family.**

       Different initial concentrations of α-pinene were used to investigate the concentration-dependent distribution of HOM
       products, summarized in Fig. 4d-f. In the gas phase, as more α-pinene reacted, more $RO_2$ was formed in the system, thus
       making self/cross reactions between two $RO_2$ more important, while decreasing the autoxidation (Eq. R1a) and unimolecular
termination (Eq. R1b). We indeed see a lower fraction of compounds with large O numbers in the gas phase when increasing
       the α-pinene concentrations (Fig. S3), while $C_8$ monomers (likely from Eq. R2b followed by fragmentation) and all dimers
       (from reaction Eq. R2c) increased (Fig. 4d). The concentration-dependent trend of HOM distributions in the gas phase is
       consistent to that observed at relatively low α-pinene conditions (17-1692 ppt) with ~30 ppb $O_3$ (Molteni et al., 2019), despite
       clearly higher α-pinene (53-120 ppb) and $O_3$ (~33 ppb) concentrations used in this work.

When adding NaCl seed particles to convert more HOM vapors into SOA, mass spectra of the condensable gas phase could
       be obtained (i.e. the difference of gas-phase measurements between without and with seed particles, Fig. 4b). Low volatile
       monomers and almost all dimers (with roughly m/z > 350 Th including the reagent ion $NO_3^-$) showed comparable, largely
       kinetically limited, and irreversible condensation loss on the seed particles, similar to results from an earlier study (Peräkylä
       et al., 2020). The mass spectra of condensed gas-phase HOM were in general comparable to those of the gas phase (Fig. 4b
vs. 4a and 4e vs. 4d), though with slightly higher dimer contribution. However, the particle-phase HOM distribution was
       dramatically different from the (condensed) gas phase measurements, and measured dimer concentrations were even higher



than monomers (Fig. 4f). With more α-pinene reacted, the enhancement of $C_{16}$-$C_{19}$ dimers in the particle phase was much larger than that observed in the gas phase, while the particle-phase O number distribution did not show obvious concentration-dependent changings among different experiments (Fig. S3). In addition, $C_{17}H_{26}O_9$ was identified in both gas

and particle phases, whereas $C_{17}H_{26}O_{7,8}$ were only detected in the particle phase. These compounds have also often been identified from filter measurements (Müller et al., 2009; Zhang et al., 2015; Mutzel et al., 2015; Kristensen et al., 2020). All these particle and gas phase comparisons raise an interesting question-- what is the source of those $C_{16}$-$C_{19}$ dimers?

The evenly distributed $C_{17}$-$C_{20}$ HOM dimers in the gas phase and their extremely low volatilities (Peräkylä et al., 2020; Hyttinen et al., 2021) would have us expecting a similar distribution in the particle phase. However, the very different HOM

distribution measured in the two phases implies that particle phase reactions are taking place. We cannot rule out that our VIA sampling setup is perturbing the true distribution, and these limitations are discussed in more detail in section 3.4. Earlier studies have speculated on different processes forming dimers in the particle phase. For example, $C_{17}H_{26}O_8$ (nominal mass at 358 amu, m/z 420 in Fig.4), which showed the highest signal in the dimer range (negligible in the gas phase), was also reported as the most abundant oligomer in α-pinene SOA measured by a High-Performance Liquid Chromatography

and Fourier Transform Ion Cyclotron Resonance-Mass Spectrometry (HPLC/FTICR-MS) (Müller et al., 2009). In that study, liquid chromatography followed by mass spectrometry showed that $C_8H_{12}O_4$ and $C_9H_{14}O_4$ (cis-pinic acid and isobaric compounds detected at m/z 185) are monomeric units that make up this $C_{17}$ dimer. It was later proposed that $C_8H_{12}O_4$ could be terpenylic acid (and isobaric compounds detected at m/z 171) (Yasmeen et al., 2010; Beck and Hoffmann, 2016). However, the proposed esterification process needs aqueous and acid conditions to turn the lactone group of terpenylic acid

into two alcohol moieties (Beck and Hoffmann, 2016), which is not available in this study (dry NaCl seed particles). In addition, terpenylic acid was proposed to be formed either through OH initiated oxidation of α-pinene or an $O_3$ initiated pathway of α-pinene oxide (yield ~2%) (Claeys et al., 2009). However, in the CO (used as an OH scavenger) experiment (No. 4, discussed in the next section), when both these pathways were very limited, $C_{17}H_{26}O_8$ was still observed as the highest dimer peak in Figure 5b. In addition, it has been shown in other studies that a dominate role of dimers (including

$C_{17}H_{26}O_8$) in the particle phase were primarily measured from ozonolysis of α-pinene (Kristensen et al., 2014; Zhang et al., 2017).

Recently, Claflin et al. (2018) used derivatization-spectrophotometric methods to quantify the distribution of different functional groups in very high-concentration α-pinene SOA (~3 mg m$^{-3}$). They compared the measurements to the modeled results of the Master Chemical Mechanism and proposed that hydroperoxides/peroxycarboxylic acids along with

aldehydes/ketones could be converted to carboxyl monomers and ester dimers through the Baeyer-Villiger reaction in SOA. This would be an explanation of those "newly formed" dimers measured in the particle phase, which may also be important under atmospheric conditions (i.e. low SOA concentrations). Furthermore, particle phase decomposition of more carbon-containing HOM compounds (e.g. $C_{10}$ in the monomer range and $C_{20}$ in the dimer range) was proposed as an important decay pathway of α-pinene SOA (Pospisilova et al., 2020), which could largely explain the relative enhancement of $C_{16}$-$C_{19}$

comparing to $C_{20}$ compounds. However, from their results, $C_{16}$-$C_{17}$ signals peak after ~2 hours of the α-pinene ozonolysis in batch-mode experiments, whereas the residence time is ~1 hour for the continuous-mode experiments conducted in this work. Nevertheless, about half of the compounds were proposed to have half-lives of less than one hour (Pospisilova et al., 2020). If the thermal energy supplied during the evaporation of SOA within the VIA can promote this decay process, then the observed particle-phase HOM distribution would be expected to shift to lower C numbers.

Although the above discussion mainly focused on $C_{17}$ compounds, $C_{18}$-$C_{20}$ HOM dimers observed in the gas phase were still identified in the particle phase. Thus, the particle phase dimers measured in this work may be partly intact HOM dimers formed and condensed from the gas phase (Berndt et al., 2018; Jokinen et al., 2014), and partly particle-phase formed dimers





which are multifunctional organic compounds being detected by a $NO_3$-CIMS (thus were regarded as "HOM"). However, as discussed in section 2.2, some part of the enhancement of $C_{16}$-$C_{19}$ compounds observed in the particle phase is likely due to a higher detection efficiency for HOM dimers than monomers with the VIA setup at 230 ℃. In order to understand this "dimer enhancement" better, we conducted experiments where we tried to modify the dimer/monomer (D/M) ratios of HOM in the gas phase, by adding CO or NO to the system (as discussed in the following sections), to investigate the effects of the corresponding D/M ratios in the particles.

### 3.1.2 α-pinene and $O_3$ dark reactions with CO

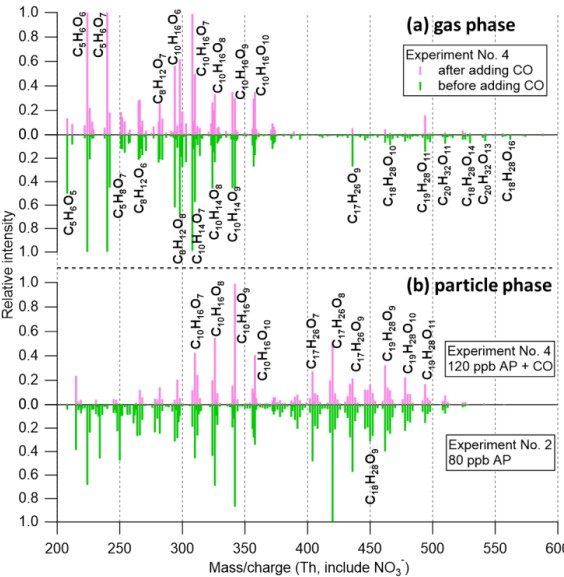

**Figure 5. Mass spectra of (a) gas phase (30-min average) measurements before (green) and after (pink) adding 100 ppm CO in experiment No.4 (input flow concentration as 120 ppb α-pinene and 33 ppb $O_3$), normalized to $(C_{10}H_{14}O_7)NO_3$ at m/z 308 with m/z 224 and m/z 240 off the scale, and (b) particle phase (60-min average) measurements with (pink) and without (green) CO, normalized to $(C_{10}H_{16}O_9)NO_3$ at m/z 342 and $(C_{17}H_{26}O_8)NO_3$ at m/z 420, respectively. Note that oxidation rate of α-pinene decreased to about half due to the decreased OH oxidation after adding CO (assuming OH yield being ~0.8 from ozonolysis of α-pinene). Therefore, experiment No. 2 (80 ppb α-pinene) was chosen for the particle-phase comparison because of comparable SOA concentrations.**

In experiment No. 4, CO was used as an OH scavenger (i.e. to isolate $O_3$ initiated reactions) to test the hypothesis that $O_3$ derived $RO_2$ and corresponding monomers are important for dimer formation in the particle phase (Kristensen et al., 2014; Zhang et al., 2017) and to change the gas-phase HOM distribution. α-pinene and $O_3$ were first added into the chamber similar to experiment No. 3, whereafter CO was added to largely shut down OH initiated formation of HOM. The added CO increases $HO_2/RO_2$ ratios compared to the pure α-pinene ozonolysis systems (roughly by a factor of $10^4$ estimated by the kinetic model), making $HO_2$ a competitive partner to react with $RO_2$ (Eq. R3a,b). Thus, lower relative concentrations of dimers were observed (Fig. 5a). Total HOM concentrations also decreased after adding CO, likely due to the decreased oxidation rate of α-pinene (Fig. S4a). However, enhancement of $C_{16}$-$C_{19}$ dimers compared to monomers was still observed in the particle phase (Fig. 5b and Fig. 4f). Furthermore, the peaks identified in the particle phase were almost the same as the ones without CO, strongly suggesting that ozonolysis products of α-pinene are important and perhaps necessary for the formation of these dimers (Kristensen et al., 2014; Zhang et al., 2017). In fact, the mass spectra of the CO experiment (No.4)

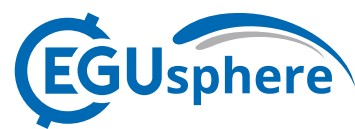

correlated well with the ones of experiment No. 2 without CO, with $r^2$ = 0.82, 0.75, and 0.80 for gas, condensed gas, and

particle phase measurements, respectively.

In addition, although the majority of HOM products decreased largely in the gas phase after adding CO, $C_{10}H_{16}O_{8,10}$ increased

significantly (Fig. S4a), which was also obvious from their time series (Fig. S5). This rapid increase with CO suggested that

$C_{10}H_{15}O_{8,10}$ existed as dominant $RO_2$ in the gas phase, and were terminated rapidly by $HO_2$ forming hydroperoxides (Eq.

R3a). On the other hand, $C_{10}H_{18}O_z$ and $C_{20}H_{34}O_z$ compounds were only formed through reactions of OH-derived $RO_2$ and

therefor decreased after adding CO. Although $C_{10}H_{14}O_z$ and $C_{20}H_{30}O_z$ compounds were formed only through ozonolysis of

α-pinene (Molteni et al., 2019), a general decrease was also observed, likely owing to the $RO_2$ autoxidation in some cases

being outcompeted by $HO_2$.

### 3.1.3 α-pinene and $O_3$ reactions with NO (UV lights on)

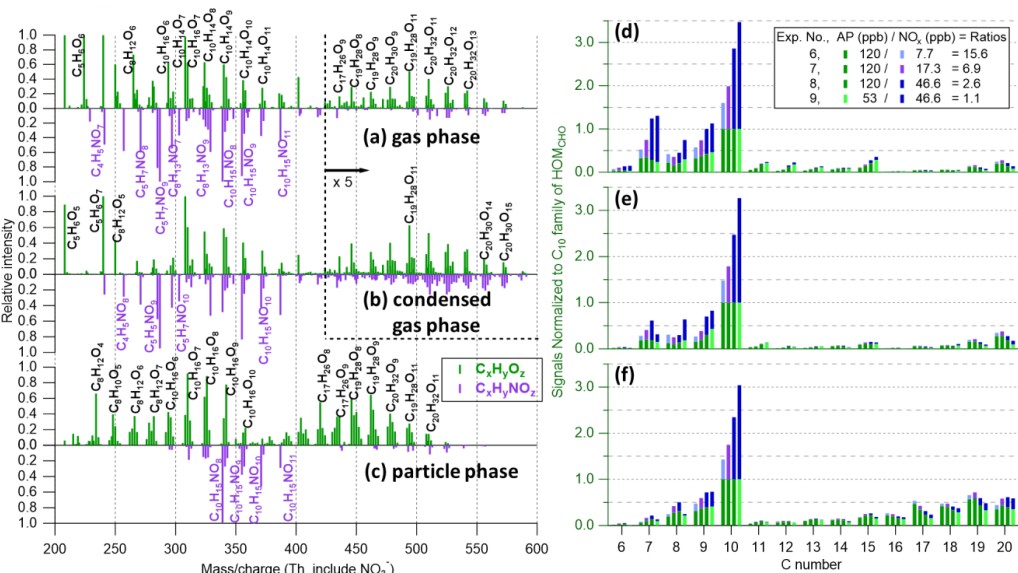

**Figure 6. (left) Mass spectra of (a) gas phase (without seed), (b) condensed gas phase (from the difference between spectra without and with seed particles), and (c) particle phase (at 230 ºC) measurements from experiment No.7 (input flow condition of 120 ppb α-pinene, 33 ppb $O_3$, and 17.3 ppb $NO_2$ with UV lights on). Gas phase and condensed gas phase mass spectra (30-min average) are normalized to $(C_{10}H_{14}O_7)NO_3$ at m/z 308 with m/z 240, and/or m/z 208 and m/z 224 off the scale, with peaks larger than 420 Th being multiplied by a factor of 5. Particle phase mass spectra (60-min average) are normalized to $(C_{10}H_{16}O_7)NO_3$ at m/z 310 with**

**m/z 339 off the scale. Nitrogen-containing ions ($C_xH_yNO_z$, i.e. HOM$_{ON}$) are plotted downward while the other ions are plotted upwards. (right) The distribution of HOM products (varying initial α-pinene to $NO_x$ ratios from 15.6 to1.1), grouped by carbon numbers, in (d) gas phase, (e) condensed gas phase, and (f) particle phase. Signals are normalized in each experiment to the $C_{10}$ family of HOM$_{CHO}$ compounds (green bars) and the blue bars (HOM$_{ON}$) stack on the top of the HOM$_{CHO}$.**

NO (generated from the photolysis of $NO_2$) was introduced into the α-pinene ozonolysis system in experiments No. 6-9, and

it acted similarly to $HO_2$ in the CO experiment to partly terminate $RO_2$ into monomers instead of forming dimers in the gas

phase. Although a relatively smaller branching ratio towards the formation of HOM$_{ON}$ monomers (Eq. R4a) than RO (Eq.

R4b) would be expected, one more N atom makes the HOM$_{ON}$ monomers (dominated by $C_{10}H_{15,17}NO_z$ and $C_9H_{13,15}NO_z$

compounds, in Fig. 6) easier to track, compared to the enhanced hydroperoxides in the CO experiment. As shown in Figure

6d, with lower initial AP/$NO_x$ ratios, the contribution of HOM$_{ON}$ monomers increased and HOM$_{CHO}$ dimers decreased in the

gas phase. Similar effects of NO-- terminating $RO_2$ and forming organic nitrates instead of dimers, thus suppressing SOA





formation-- were reported from previous studies (Ehn et al., 2014; Zhao et al., 2018a; Pullinen et al., 2020). The condensed gas phase followed the patterns of the gas-phase measurements.

Particle-phase HOM again showed relatively higher contributions of dimers ($C_{16}$-$C_{20}$ compounds) compared to the gas phase (Fig. 6f vs. 6d). However, different from experiments No. 1-4, NO was used in experiments No. 6-9 to terminate $RO_2$ to

$RONO_2$, which are not as effective particle-phase precursors as $HOM_{CHO}$ monomers to form dimers through the Baeyer-Villiger reaction in SOA (Claflin et al., 2018). Consequently, less "enhancement" of $C_{16}$-$C_{19}$ $HOM_{CHO}$ compounds in the particle phase was observed compared to experiments without $NO_x$ (Fig. 6f vs. 4f). In addition, highly functionalized organic nitrates observed in the particle phase have been found to decompose within 2-4 hours based on ambient measurements (Lee et al., 2016), suggesting that the formation of $HOM_{ON}$ might be larger than we detected.

**3.1.4 α-pinene and $NO_3$ dark reactions (UV lights off)**

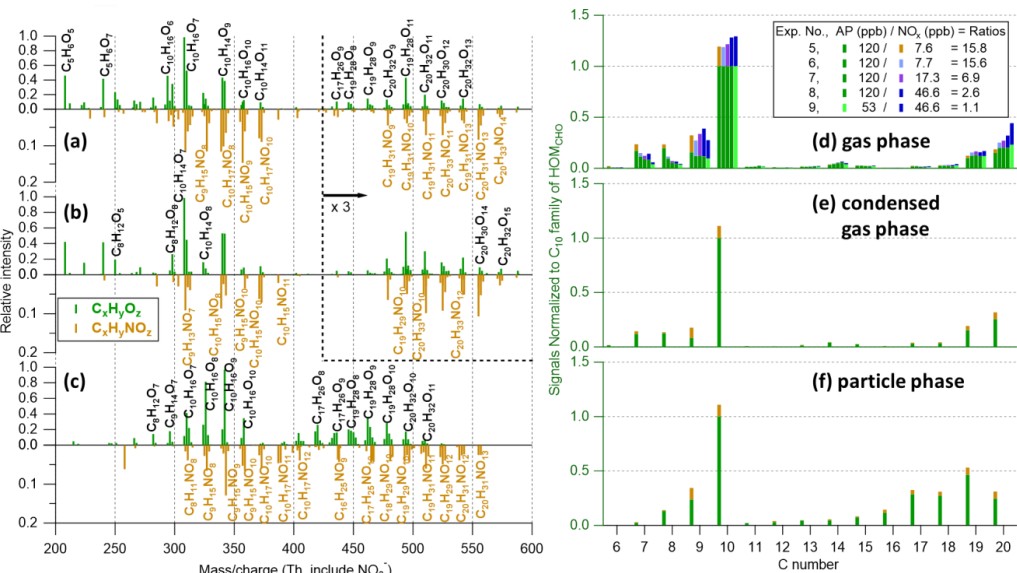

**Figure 7.** (left) Mass spectra of (a) gas phase (without seed), (b) condensed gas phase (from the difference between spectra without and with seed particles), and (c) particle phase (at 230 ºC) measurements from experiment No.5 (input flow condition of 120 ppb α-pinene, 33 ppb $O_3$, and 7.6 ppb $NO_2$ with UV lights off). Note that the y-axis of $HOM_{CHO}$ (green spectrum, upward) and $HOM_{ON}$

(brown spectrum, downward) are on different scales. Gas phase and condensed gas phase mass spectra (30-min average) are normalized to $(C_{10}H_{14}O_7)NO_3$ at m/z 308, with peaks larger than 420 Th being multiplied by a factor of 3. Particle phase mass spectra (60-min average) are normalized to $(C_{10}H_{16}O_9)NO_3$ at m/z 342. (right) The distribution of HOM products (varying initial α-pinene to $NO_x$ ratios from 15.8 to1.1), grouped by carbon numbers, in (d) gas phase, (e) condensed gas phase, and (f) particle phase. Signals are normalized to $HOM_{CHO}$ $C_{10}$ family. The UV lights were off during the entire experiment, making experiment

No.5 ($HOM_{ON}$ as brown bars) the only one with particle-phase HOM measurements from α-pinene and $NO_3$ reactions.

Particle-phase HOM from $NO_3$ initiated oxidation was investigated by adding $NO_2$ into the α-pinene ozonolysis system under dark conditions (i.e. UV lights off) in experiment No. 5. $NO_3$ oxidation of alkenes almost exclusively undergoes electrophilic addition to the double bond instead of H abstraction (Kerdouci et al., 2010), thus multigeneration oxidation by $NO_3$/$O_3$ should be negligible. In experiments No. 6-9, α-pinene, $O_3$, and $NO_2$ (7.6-46.6 ppb) were added into the chamber

until a steady state was reached, whereafter the UV lights were turned on to photolyze $NO_2$ and $NO_3$ to NO (results discussed in section 3.1.3). Thus, only gas-phase measurements from $NO_3$ initiated oxidation were obtained from those experiments



(Fig. 7d), whereas experiment No. 5 was conducted with UV lights off during the entire period, making it the only with particle-phase HOM measurements of the α-pinene and $NO_3$ reactions (Fig. 7f).

For both NO and $NO_3$ experiments, $HOM_{ON}$ were dominated by compounds with only one nitrogen atom (94.5+0.9%) in both gas and particle phases. This result indicates that NO-termination of $NO_3$-derived nitrooxy-$RO_2$ was negligible in both systems, and likewise that HOM dimer formation through cross-reactions of two $nRO_2$ were unlikely. Note that many previous studies used $N_2O_5$ as the $NO_3$ source (and often with concentrations higher than VOCs) to avoid $O_3$ initiated oxidation, thus $C_xH_yN_2O_z$ compounds (e.g. $C_{20}H_{32}N_2O_{9,10}$) were the largest signals in the dimer range, formed by self/cross reactions of two $nRO_2$ (Claflin and Ziemann, 2018; Takeuchi and Ng, 2019; Bell et al., 2021). Here, $O_3$ derived $RO_2$ still contributed largely to HOM in this work, thus cross reactions of $nRO_2$ with $RO_2$ was the main process forming dimers (with only one N-atom, e.g. $C_{19}H_{31}NO_z$ and $C_{20}H_{31}NO_z$). In addition, since $HOM_{ON}$ signals generally increased after turning on the UV lights (Fig. S6), peroxy acyl nitrates (PANs) formed through $NO_2$-termination of $RO_2$ (Rissanen, 2018) were likely less important than NO-termination due to lower reaction rates and lower thermal stability of the products (Atkinson, 2000; Orlando and Tyndall, 2012; Berndt et al., 2015).

In the $NO_3$ experiment (experiment No.5, Fig. 7), $C_{10}$ compounds dominated $HOM_{CHO}$ in the monomer range in both gas and particle phases, while $C_9$ and $C_{10}$ compounds were the largest $HOM_{ON}$ monomers. Similarly, $C_{10}H_{15,17}NO_z$ and $C_9H_{13,15}NO_z$ were identified as the main $HOM_{ON}$ products during $NO_3$ initiated α-/β-pinene oxidation using a FIGAERO-iodide-CIMS for gas- and particle-phase measurements (Boyd et al., 2015; Nah et al., 2016; Takeuchi and Ng, 2019) and when using an EESI-TOF for SOA measurements at the Hyytiälä boreal forest site (Pospisilova et al., 2020). A detailed comparison of $HOM_{ON}$ peaks identified in the particle phase from this work with other techniques is summarized in Table S3. Again, $C_{16}$-$C_{18}$ compounds were nearly exclusively measured in the particle phase along with a relative enhancement of $C_{19}$ compounds compared to the gas phase, indicating potential particle-phase related processes.

### 3.1.5 Summary of dimer/monomer distributions

In this section, the dimer to monomer (D/M) ratio was calculated for each experiment to relate the distribution of HOM products in both phases. Overall, D/M ratios observed in the particle phase followed the trend of gas phase for all experimental conditions. As dimers condensed more readily than monomers, the gas phase D/M ratios decreased after adding seed particles (Fig. S7d) and the condensed gas phase D/M ratios (grey markers in Fig. 8b) was roughly 1.5-fold compared to the gas phase. However, the D/M ratios measured in the particle phase were 2-9 times as high as those observed in the (condensed) gas phase (Fig. S8). Interestingly, the experiments where we observed the lowest gas-phase D/M ratios (experiment No. 4, 8, 9) were the ones where the particle-phase D/M enhancement was the largest. In case there was dimerization taking place in the particles, this process should indeed be seen most clearly when the amount of condensing dimers was the lowest. Overall, these highest discrepancies between different phases are larger than the expected underestimation of monomer signals (by a factor of ~2 estimated by comparing the measurements between 230 ºC and 170 ºC, as shown in Fig. 2c for $C_{10}$ compounds). Nevertheless, further investigations will be needed to study to what extent could this enhancement be attributed to particle-phase chemistry.

One striking feature of Fig. 8b is that despite the D/M ratio of gas-phase HOM varying by a factor of 3 between different experiments (i.e., range of y-values for red markers in Fig. 8b), the relative changes of D/M ratios between different experiments (for the same phase) were clearly lower. For example, between experiments No. 1 and No. 3, the gas phase D/M ratio increased by ~50 % (from 0.2 to 0.3) and the particle phase D/M ratio increased by a very similar amount, ~54 % (from 1.1 to 1.7). We can only speculate about the underlying reasons behind this behavior. One possibility is that there is dimer formation in the particles that scales suitably to produce the observed results. Another possibility is that our sensitivity





to HOM monomers in the particle phase is lower than for HOM monomers in the gas phase. As indicated earlier, the average C-atom content in the HOM dimers was lower in the particles than in the gas phase under all conditions. The loss of C atoms is unlikely to be limited to the dimers, and similar changes in the monomers can be expected (whether through particle-phase chemistry or thermal decomposition). The loss of a few functional groups from a HOM monomer can potentially make it undetectable by the $NO_3$-CIMS, which in turn would cause an increase in the observed D/M ratio. Future studies will hopefully be able to shed more lights on this matter. The potential role of thermal decomposition upon heating cannot be ruled out and will be discussed further in section 3.4.

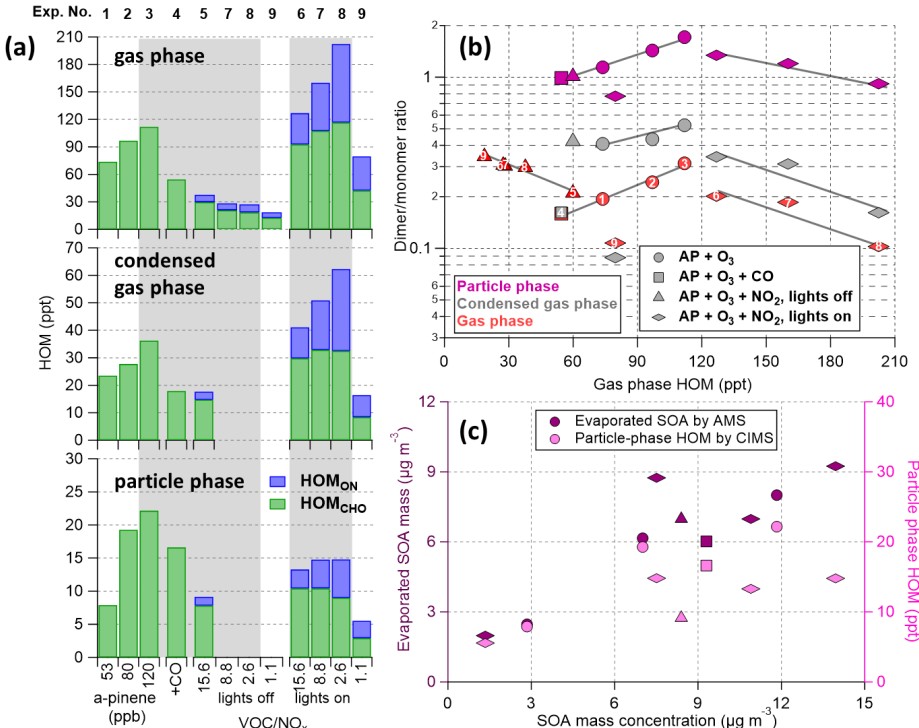

**Figure 8.** Summary of (a) total HOM concentrations estimated in the gas, condensed gas, and particle phase (different scales were used for different phases), (b) dimer to monomer (D/M, i.e. $C_{16}$-$C_{20}$/$C_8$-$C_{10}$) ratios, and (c) particle phase HOM and SOA measurements. The evaporated SOA mass was calculated from AMS measurements (i.e. the mass difference between VIA and bypass modes). Experiment numbers are given in panel (a) and (b). Shaded areas in panel (a) represents experiments with initial α-pinene concentrations of 120 ppb. The HOM concentrations are given in absolute values here to be able to compare the different experiments more quantitatively, but the large uncertainties (at least a factor 3) need to be kept in mind. The gas-phase HOM concentration is used as the x-axis in (b) as a surrogate for the overall α-pinene reaction rate, which could not be plotted directly due to the lack of VOC measurement. Grey solid lines in panels (b) only serve as guides to connect experiments with similar conditions.

In the NO experiments (No. 6-9 with UV lights on), D/M ratios of the condensed gas phase HOM changed the most, nearly by a factor 4 (Fig. 8b). However, the concurrent decrease in particle phase D/M ratios was smaller, closer to a factor 2. This change is also clear from Fig. S8. If there was particle phase dimer formation taking place, the effect should be most visible when the gas phase dimer contributions are the lowest, as in these experiments. While the very high $NO_x$/VOC ratios in these experiments make the direct comparison to other experiments hard, this can be seen as an indication that particle phase dimer formation is indeed taking place, as has been suggested earlier (Kalberer et al., 2004; Claflin et al., 2018). It should also be noted that in these experiments, the gas and particle phase absolute concentrations changed clearly from other





experiments (Fig. 8a). HOM concentrations (= $HOM_{CHO}$ + $HOM_{ON}$) increased 13-81% in the gas phase, whereas particle-phase HOM decreased 33-40% (and SOA decreased 11-50%, Fig. S7) compared to the reference α-pinene and $O_3$ experiments (No. 1 and 3). This indicates a higher average volatility of the observed gas phase HOM, largely explainable by suppressed formation of low volatile $HOM_{CHO}$ dimers and increased formation of more volatile $HOM_{ON}$ monomers.

Exactly how this may impact the D/M ratios in gas and particle phase HOM remains an open question.

As there are still uncertainties relating to the VIA response, e.g. in correctly accounting for wall losses of vapors in the current experimental setup (especially within the VIA inlet), only a qualitative comparison was given here. As shown in Fig. 8c, ~80% of SOA evaporated within the VIA system based on AMS measurements, which is consistent with the integrated mass based on SMPS datasets (Fig. S7b). However, particle-phase HOM concentrations measured by the VIA-NO$_3$-CIMS

was much lower (by a factor of ~5-6, accounting for the dilution and assuming an average molecular mass of HOM compounds as 300 g mol$^{-1}$) compared to the AMS measurements. This difference is large enough that even after accounting for our large uncertainties in quantification, we expect that a noticeable fraction of the particle-phase products are not detect using this system. Vapor losses to the walls after evaporation as well as the selectivity of our ionization are expected to be the two main reasons for this. More detailed characterization of this inlet system will be investigated in future studies in

order to evaluate if the wall losses can either be avoided or accounted for. Nevertheless, the trend of particle-phase HOM is comparable to the SOA measurements (Fig. 8c), with the largest discrepancies observed under high SOA loadings with NO$_x$ added.

**3.2 Elemental composition of HOM**

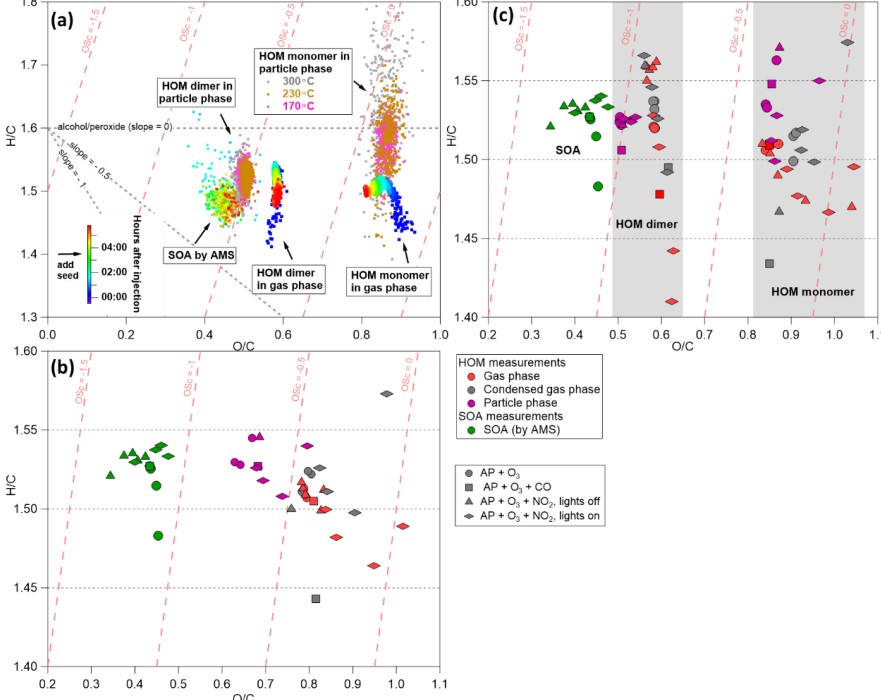

**Figure 9. Van Krevelen diagrams of the VIA-NO$_3$-CIMS and AMS data. (a) Gas-phase HOM (10-s averaged data points) and SOA measured by AMS (20-s averaged data points) from experiment No. 1. The SOA and the gas-phase HOM data are colored**



**by the time after α-pinene and O₃ injection, while the particle-phase HOM are colored by the temperature of the VIA. (b) Average "bulk" HOM and SOA elemental ratios for all experiments. (c) Average gas phase (red, 30-min), condensed gas phase (grey, 30-min), and particle phase (purple, 60-min) HOM elemental ratio separated into monomers (C₈-C₁₀) and dimers (C₁₆-C₂₀).**

In this section, Van Krevelen diagrams are used to investigate the bulk chemical properties of HOM measured in both gas and particle phases, and compared to the more routine elemental analysis from the AMS (Decarlo et al., 2006; Aiken et al., 2007; Aiken et al., 2008; Canagaratna et al., 2015).

As shown in Fig. 9a, gas-phase HOM monomers formed rapidly after the injection of α-pinene and O₃, with bulk chemical composition stabilizing at elemental ratios resembling $C_{10}H_{15}O_8$. Dimers followed the H/C of monomers but showed lower

O/C ratios, as has been observed also in earlier studies (Jokinen et al., 2014; Berndt et al., 2018). In the particle phase, the chemical composition of HOM measured at different temperatures were quite comparable, thus the datasets obtained at 230 ºC were used to represent the particle-phase HOM measurements. Since the elemental ratios of HOM measurements are relatively stable 1.5 hours after injection of gas precursors, signal-weighted means were calculated for each phase after reaching a steady state for each experiment separately (Fig. 9b, c). In general, it is interesting that all data sets showed

comparable H/C ratios (1.5±0.1) but a relatively large range of O/C ratios. Especially, O/C ratios of HOM measured in the particle phase (purple markers in Fig. 9b) are lower compared to the gas (red markers) and condensed gas (grey markers) phase measurements (0.63-0.80 vs. 0.76-1.02). The AMS measured SOA showed the lowest O/C ratios (0.34-0.48), which are comparable with previous studies (Alfarra et al., 2006; Chhabra et al., 2010), summarized in Table S4. One possible explanation for the inconsistence of O/C ratios between AMS and CIMS measurements is that the VIA-NO₃-CIMS measured

only the most oxidized part of the SOA (as discussed in section 3.1.5).

HOM measurements were also separated into monomers and dimers for more detailed comparisons (Fig. 9c). In α-pinene ozonolysis experiments, the chemical compositions of gas-phase HOM are comparable under different initial α-pinene concentrations (red circles in Fig. 9c). The condensed gas phase monomers showed slightly higher O/C ratios, while dimers showed comparable O/C ratios compared to the gas phase (red vs. grey circles), which is consistent with the volatility

distribution of HOM products (Peräkylä et al., 2020). However, a small O/C gap was observed between the particle phase and the (condensed) gas phase for dimers (0.50-0.51 vs. 0.58-0.59). In experiment No.4, adding CO largely shut down OH derived RO₂, which lead to lower H/C ratios of gas-phase dimers compared to that before CO addition (Fig. S8a). On average, the H/C ratios of dimers decreased from 1.52 to 1.48 in the gas phase and decreased from 1.55 to 1.51 in the particle phase. The AMS measurements did not show detectable effects after adding CO. Again, the gas and particle phases showed a small

gap in the dimer range (red vs. purple square in Fig. 9c) as in experiment No. 1-3.

In NOₓ experiments, H/C ratios of both gas-phase HOM monomer and dimer decreased after turning on the UV lights (Fig. S8b), owing to the interruption of NO₃ initiated oxidation (i.e. addition of NO₃ formed nRO₂ radicals have one more H atom than O₃ derived RO₂). However, all particle-phase measurements (purple diamonds and triangles in Fig. 9c) were relatively comparable with other experiments (especially for dimers), indicating similar particle-phase transformations after the

condensation of HOM from the gas phase. On the other hand, in both NO₃ (lights off) and NO (lights on) experiments, higher O/C ratios were observed in the monomer range compared to experiments without NOₓ because of three more O atoms from the nitrate functional group. In addition, higher N/C ratios were observed in both the gas (0.037-0.07 vs. 0.018-0.02) and particle phase (0.01-0.04 vs. 0.01) with UV lights on compared to those without UV lights, confirming that more HOM_ON was formed through NO and RO₂ reactions than NO₃ oxidations. Correspondingly, higher O/C ratios of SOA were observed

with UV lights on compared to the dark NO₃ experiment (0.40-0.48 vs. 0.34-0.41).



**3.3 Organic nitrate (ON) measurements**

Finally, organic nitrate measured by the AMS and the VIA-NO$_3$-CIMS system were compared to evaluate the performance of this novel system. Based on AMS measurements, nitrate mass tracked well with organics (Fig. 10a, b) for each experiment separately, indicating nitrate signals were from organic nitrates. The nitrate/organic ratio in the AMS also increase very

clearly as the NO$_x$/VOC ratio in the chamber increased. The nitrate/organic mass ratios increased from 0.012-0.061 to 0.018-0.075 after turning on the UV lights, in agreement with earlier discussion that more intensive HOM$_{ON}$ formation took place in NO experiments than in NO$_3$ experiments. The NO$^+$/NO$_2^+$ ratios, a marker for the contribution of organic nitrates to the measured nitrate signal, observed in experiment No. 6-9 did not show discrepancies between UV lights on and off (Fig. 10c), and were well separated from the data set of ammonium nitrate calibration (5.2-5.6 vs. 1.6).

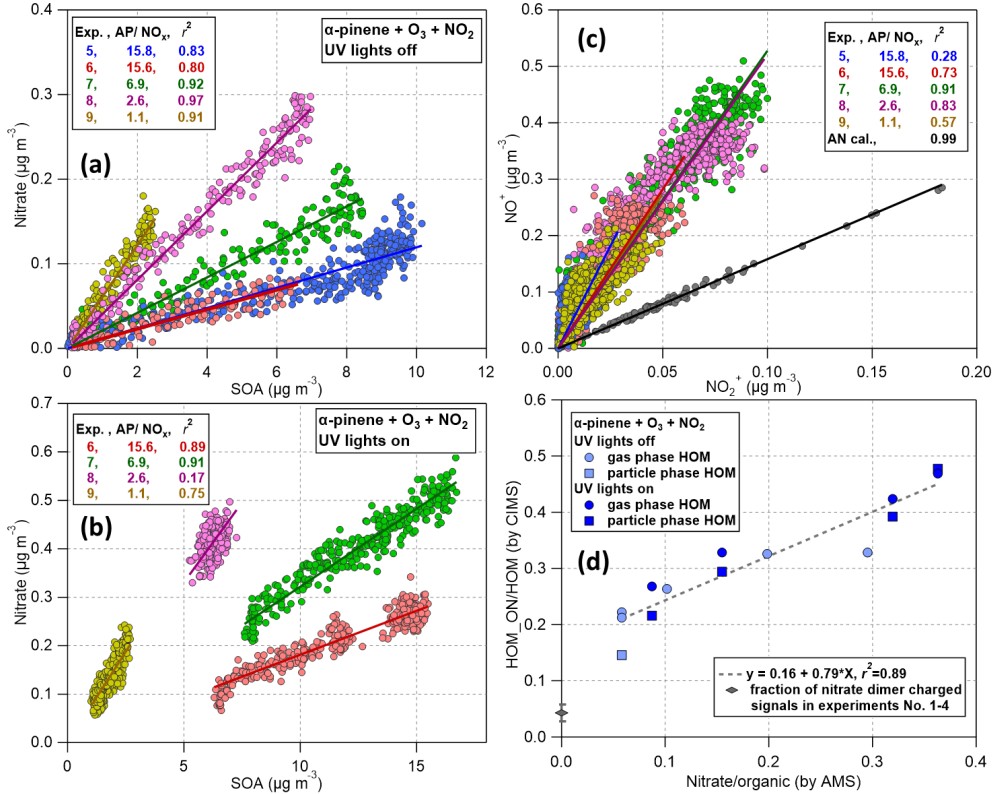


**Figure 10. Scatter plots of organic nitrate observations with the VIA-NO$_3$-CIMS and AMS. (a) AMS-measured nitrate mass vs. organic mass during NOx experiments (No. 5-9) with UV lights off. (b) Same as (a), but with UV lights on. (c) NO$^+$ vs. NO$_2^+$ ions measured by the AMS, including experiments with both UV lights off and on. (d) Molar fraction of organic nitrate (HOM$_{ON}$) to total organic signals (HOM = HOM$_{CHO}$ + HOM$_{ON}$) measured by the VIA-NO$_3$-CIMS system vs. the ratio of nitrate to organics**

**measured by the AMS ((nitrate/62)/(SOA/300)).**

Assuming an average molar mass as 300 g mol$^{-1}$ for organic molecules, the molar fraction of ON to total organic aerosols (= (nitrate/62)/(SOA/300)) was estimated to be 5.8-30% for NO$_3$ experiments and 8.7-36% for NO experiments based on AMS measurements. Interestingly, the molar fraction of HOM$_{ON}$ to total HOM signals measured in both the gas (circles in Fig. 10d) and particle phase (squares in Fig. 10d) by the VIA-NO$_3$-CIMS correlated well with the above AMS measurements,

suggesting that chemical composition of gas- and particle-phase HOM (i.e. relative contribution of HOM$_{CHO}$ and HOM$_{ON}$)



were quite comparable. Note that the slope of the fitted line is sensitive to several factors, including different calibration factors between $HOM_{CHO}$ and $HOM_{ON}$ and the estimated average molar mass of SOA. Thus, the value should be only qualitatively interpreted. We also note that some part of the non-zero intercept in Figure 10d (16%) is likely contributed by charging in the $NO_3$-CIMS by the nitrate dimer ($HNO_3 \cdot NO_3^-$), which is higher than that in α-pinene ozonolysis experiments

(estimated to be 4.3+1.5%). It is possible that this ratio would increase further in $NO_x$ experiments if there was $HNO_3$ formed at some amounts in the gas phase. Finally, neither the $NO^+/NO_2^+$ nor the ON molar fraction analysis showed any clear differences between the NO and $NO_3$ experiments based on both AMS and $NO_3$-CIMS datasets. Thus, other parameters (e.g. UV lights on/off in the chamber or hour of the day in the atmosphere) is needed to distinguish $HOM_{ON}$ formed by $NO_3$ oxidation from those by NO termination of $RO_2$.

**3.4 Discussion on the factors that may affect the HOM detection**

Thermal decomposition is a well-known issue with techniques based on the evaporation of molecules from the particle phase to identify the chemical composition of vapors, especially for labile organic mixtures. It remains unclear to what extent the thermal decomposition of SOA happened under different temperatures and/or if the thermal desorption techniques could promote particle-phase reactions. However, if thermal decomposition did play an important role in this VIA-$NO_3$-CIMS

system, we would expect more decomposition of dimers (labile RO-OR compounds) than monomers, and in particular at the highest temperature. Yet, we observed much higher dimer/monomer ratios in the particle phase than the gas phase, suggesting that we might instead "enhance" dimers if the thermal decomposition would primarily remove monomers from our sensitivity window.

In section 3.2, lower O/C ratios of the particle-phase HOM were observed comparing to the gas-phase HOM, which could

partly be attributed to the thermal decomposition of organic compounds within the VIA. The carbonyl, peroxide, esters, carboxyl, and hydroxyl were ubiquitous functional groups identified in α-pinene SOA under various conditions (Claflin et al., 2018). Decarboxylation of C(=O)O- functional groups may play a relatively more important role than the dehydration of hydroxyl (with $H_2O$ being the main thermal decomposition product) owing to their expected higher contributions to SOA. Concurrent formation of CO, $H_2O$, and $CO_2$ from decarboxylation would decrease both the O/C and H/C ratios for the

particle-phase measurements, which was observed at 200 °C on the vaporizer of the AMS (Canagaratna et al., 2015) and in the FIGAERO-CIMS (Stark et al., 2017). With proper combinations of C(=O)OH and C(=O)OR thermal decomposition, it is possible to decrease O/C as well as slightly increase H/C (e.g. on average losing more $CO_2$/CO than $H_2O$).

In addition, thermal decomposition of peroxide oligomers was observed in the thermogram of an acetate-FIGAERO measurements, with an estimated dissociation temperature around 80 °C for O-O bond (Lopez-Hilfiker et al., 2015).

However, large amounts of $C_{16}$-$C_{19}$ dimers measured in this work suggested either these labile peroxides transferred to other more stable functional groups in SOA, or the much short residence time at elevated temperature in the VIA (less than 0.2 s), as well as sampling without surface contact, may lead to less thermal decomposition compared to the traditional collection-and-desorption methods (Häkkinen et al., 2022). Nevertheless, it is clear that we need further investigations to determine if particle-phase processing or thermal decomposition (or other hardware effects) are the larger causes for the changes observed

between the particle and (condensed) gas phase in our work. In particular, the temperature-dependent sensitivity (including evaporation and transmission efficiencies) for different types of organic molecules will need to be characterized in future works.



## 4 Conclusion

Combined gas- and particle-phase HOM measurements were conducted using the novel VIA inlet combined with a $NO_3$-
CIMS in this study. α-pinene ozonolysis was studied under different conditions, including using CO as an OH scavenger
and varying $NO_x$ concentrations with and without UV lights to perturb the oxidation chemistry and gas-phase dimer
formation. Within the 50-min residence time of the chamber, we observed distinct differences between HOM in particles
and the HOM that were condensing. In particular, we noted clearly enhanced $C_{16}$-$C_{19}$ HOM dimers in SOA that were not
observed in the gas phase. This "enhancement" might be partly attributed to a slightly higher sensitivity towards dimers than
monomers using the VIA inlet system. However, different chemical composition identified between two phases implies the
existence of potential particle-phase chemistry. In addition, gas-phase dimer formation was considerably suppressed in both
the CO and NO experiments, and this also led to decreases in particle-phase dimers, yet dimers still made up a considerable
fraction of the observed SOA.

In addition to the generally shorter carbon skeletons of the particle phase dimers (mainly $C_{16}$-$C_{19}$) compared to the gas phase
(mainly $C_{19}$-$C_{20}$), average O/C ratios of the HOM (especially in the dimer range) also decreased slightly in the particle phase.
Earlier studies have proposed that e.g. Baeyer-Villiger reactions could convert peroxycarboxylic acids and hydroperoxides
to ester dimers in reactions with aldehydes/ketones (Claflin et al., 2018; Bakker-Arkema and Ziemann, 2020), which is not
inconsistent with the results observed in this study. Our results indicate that particle-phase reactions are taking place under
the time scales probed here, with some of the reactions potentially leading to the formation of new dimers, while some
reactions may be causing loss of small fragments (e.g. $C_{20}$ dimers forming $C_{17}$ dimers). These predominant $C_{17}H_{26}O_z$
compounds identified in the particle phase, which have often been reported by previous offline measurements, might be
related to HOM dimer formation in both the gas and particle phases. However, the new VIA-$NO_3$-CIMS system used in this
work will require more detailed characterization in order to better understand how the thermal desorption and wall effects
may modify the HOM distributions from those existing in the particles. The $NO_3$-CIMS is known to be selective towards
the most oxygenated molecules, which also means that we are likely not detecting all the evaporating compounds. This is
also indicated by the measured O/C ratios being clearly higher than those measured by an Aerodyne AMS. However, the
measured organic nitrate fraction measured by the VIA-$NO_3$-CIMS was found to track those measured by the AMS. Taken
together, we believe that this system is a promising technique for combined online gas- and particle-phase HOM
measurements.

*Data availability*. Data are available upon request from the corresponding authors.

*Author contributions*. ME and JZ designed the study. ME, JK, JEK, MC, and DR support the experiment setup and analysis
methods. JEK built the first version of the VIA. JZ, EH, and FG conducted the experiments. JZ analyzed the data. JZ prepared
the manuscript with contributions from all co-authors.

*Competing interests*. Jordan Krechmer, Manjula Canagaratna, and Douglas Worsnop work for Aerodyne Research, Inc.,
which developed and commercialized the VIA inlet used in this study.

*Acknowledgements*. This work was supported by funding from Academy of Finland (grants 317380, 320094, 325656,
345982, and 346370) and University of Helsinki 3-year grant (75284132). Jian Zhao thanks Lauri Franzon and Otso Peräkylä
for helpful discussions about the experiments and results. Ella Häkkinen thanks the Vilho, Yrjö and Kalle Väisälä Foundation
for financial support. Frans Graeffe acknowledge Svenska Kulturfonden (grants 167344 and 177923) for financial support.




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
