# Peer review of "A Combined Gas- and Particle-phase Analysis of Highly Oxygenated Organic Molecules (HOM) from α-pinene Ozonolysis"

_EGUsphere, 2022_

## Author Comment (AC1)

**Response to Reviewer #1**

**General comment**

The authors present an interesting piece of work, regarding the partitioning of HOM (highly oxygenated organic molecules) and formation and fate of HOM in the particulate phase. With their setup they tried to compare the loss of HOM from the gas phase (what they somewhat unfortunately call condensed phase) with those they could detect in the particulate phase. For both phase they utilized $NO_3$-CIMS, equipped with a new VIA inlet for measurement of the particulate phase (Vocus Inlet for Aerosols), though. The VIA inlet essentially uses thermal evaporation to make the SOA components available to the mass spectrometer. Within their Teflon flow chamber the authors addressed a range of different chemical regimes with the purpose to achieve a range of products distributions and to verify particulate observations in response to the change of the actual chemical regime. Unfortunately, it is not clear (yet) how far the observations in particulate late phase are methodologically biased. Here the authors suffer from the usual problems of thermos evaporation and the quantitative transfer in into the mass spectrometer (and other instruments). Insofar it is not quite clear to me in how far we look at interesting and new results in SOA formation or at interesting artefacts of baking and loss of SOA components. I find also the modelling efforts a bit simple, but ok, they are not in center of the manuscript.

Nevertheless, I find that the results of the study are very interesting, and what is more important: the courageous approach is inspiring. I see to initiate discussions and new approaches also as part of good science. Of course, the authors did efforts to characterize the limits of their methods and very positively they discuss the limits quite openly and self-critical. They speculate but never do improper claims. Moreover, the paper is well written and well structured. The material is presented clearly and in a suited manor. The figures are dense however after some looking at them keep the important information together in one place.

Having all limitations in mind, I would still say this is a quite excellent piece of work. Therefore, I suggest to publishing the manuscript as it is in ACP.

The authors may consider my suggestions for slight improvements.

We thank the reviewer for the positive and insightful comments, and we answer the specific comments point-by-point below. The reviewer's comments are in **blue**, and our answers are in **black**.

**Specific comments**

**Comment #1:**

I have questions regarding the shift from $RO_2$ regime to $HO_2$ regime:

Line 391/397: If you shift from $RO_2$ to $HO_2$ regime, wouldn't you expect that the different set of termination products in the gas-phase – more hydroperoxides - should affect also the particulate chemistry?

**Response:**

Yes. If the Baeyer-Villiger reaction played an important role in the particle phase to form dimers, then enhanced gas-phase formation of hydroperoxides, as the key precursors of the Baeyer-Villiger reaction, would result in enhanced formation of dimers in SOA. This hypothesis is supported by the experiment with CO addition— although dimers decreased largely in the gas phase, we still observed quite high dimer signals in the particle phase.

However, the changes in the chemical composition of the particle-phase HOM were not as significant as we expected (at least not as the changes in concentration mentioned above). One possible explanation might be that there was already a large amount of hydroperoxides condensed in the $RO_2$ regime. Converting more $RO_2$ to hydroperoxide monomers instead of forming ROOR dimers in the gas phase, would lead to enhanced condensation of hydroperoxides. These monomers have similar structures (and chemical properties) as the products from pure α-pinene ozonolysis. Thus, a higher concentration of hydroperoxides might mainly increase the formation rate of dimers and through similar pathways. Consequently, they end up as comparable dimer products.

**Comment #2:**

Line 400: The shift to more $C_{10}H_{16}$ compounds alone does not mean that autoxidation is hold by $HO_2$. Were the $C_{10}H_{16}$ compounds on average less oxidized than the $C_{10}H_{14}$ compounds they replaced?

**Response:**

The reviewer is correct that "the shift to more $C_{10}H_{16}$ compounds alone does not mean that autoxidation is hold by $HO_2$". Instead of the argument we made— "the $RO_2$ autoxidation in some cases being outcompeted by $HO_2$", it would be more appropriate to mention that "the $RO_2$ autoxidation and $RO_2$ cross-/self- reaction were shifted towards $RO_2$ and $HO_2$ termination reaction".

The decreased $RO_2$ autoxidation is supported by the results in Figure S5 and Figure R1, showing that $C_{10}H_{14}O_{>7}$ compounds are in general less oxidized with CO addition. The decreased $RO_2$ cross-/self-reaction is supported by the results in Figure S5 and Figure 5, showing that the gas-phase dimers were largely decreased with CO addition.

[Figure]

**Figure R1. $C_{10}H_{14}O_z$ and $C_{10}H_{16}O_z$ compounds measured before and after CO addition in the gas phase.**

However, if the "newly" formed $C_{10}H_{16}$ compounds are less oxidized than the "old" $C_{10}H_{14}$ compounds is complicated, since $C_{10}H_{16}$ compounds can be formed from both OH- and $O_3$-derived $RO_2$ through various uni-/bi-molecular pathways. 1) The increase of $C_{10}H_{16}O_{8,10,12}$ suggested that they might be formed through $HO_2$ and $C_{10}H_{15}O_{8,10,12}$, in particular the rapid increase of $C_{10}H_{16}O_{8,10}$ (Figure S5).

Otherwise, we would have observed decreases of $C_{10}H_{16}O_{8,10,12}$ if OH-derived $C_{10}H_{17}O_{7,9,11}$ dominated their formation. 2) $C_{10}H_{16}O_{7,9,11}$ decreased after CO addition, suggesting that either $O_3$ related $RO_2$-$RO_2$ pathway and/or OH related RO (carbonyl channel of two $C_{10}H_{17}O_{8,10}$ radicals) decreased. Thus, the average oxidation state of these $C_{10}H_{16}$ compounds with CO might be roughly comparable to the $C_{10}H_{16}$ compounds without CO and thus might be even more oxidized than $C_{10}H_{14}$ compounds (they are less oxidized). But there might be a point if we introduce an excess amount of $HO_2$, the entire $RO_2$ autoxidation could be really inhibited.

Overall, to make this point clear, we modified this argument to:

*"Although $C_{10}H_{14}O_z$ and $C_{20}H_{30}O_z$ compounds were formed only through ozonolysis of α-pinene (Molteni et al., 2019), a general decrease was also observed, likely owing to the shift from $RO_2$ regime (i.e. $RO_2$ autoxidation and $RO_2$ cross-/self- reaction) to $HO_2$ regime (i.e. $RO_2$ and $HO_2$ termination reaction). The decreased $RO_2$ autoxidation is supported by the results in Figure S5, showing that $C_{10}H_{14}O_{>7}$ compounds are in general less oxidized with CO addition. The decreased $RO_2$ cross-/self-reaction is supported by the results in Figure S5 and Figure 5, showing that the gas-phase dimers were largely decreased with CO addition."*

**Typos etc.**

**Comment #3:**

Line 43: I suggest to replacing "largest" by "most abundant" or so.

**Response:**

We changed "largest" to "most abundant" in the manuscript.

**Comment #4:**

Line 141ff: It is not clear if you use LTOF or HTOF MS. Are you connecting to an Eisele Inlet?

**Response:**

Yes, we used an Eisele Inlet and an LTOF MS. In order to make this clear to the readers, we added the following sentence in line 141: *"The $NO_3$-CIMS was equipped with an Eisele Inlet (Eisele and Tanner, 1993) and a long time-of-flight mass spectrometer, providing a mass resolution of ~8500 above 125 Th."*

**Comment #5 & #6:**

Figure 2c: Were the raw signals normalized to reagent ions or total ion count. If not, why not?

Figure 2d: One could sacrifice the same scale as for the SMPS data and enlarge the data in right hand panel. Finally, you refer to details in this panel in the text.

**Response:**

The raw signals were not normalized to reagent ions or total ion counts. Although the signal of reagent ions was quite stable, we should have normalized the particle-phase measurements to the reagent ions. Thus, we made corresponding changes in Figure 2c. Also, we modified Figure 2d following the reviewer's suggestion. The updated Figure 2 is attached here and replaced the one used in the manuscript.

[Figure]

*"Figure 2. Overview of experiment No. 1 (input flow with 53 ppb α-pinene and 33 ppb O₃). (a) Particle number concentration and size distribution sampled by SMPS as a function of time. Time series of (b) measured and modeled ozone and α-pinene concentration (1-min averaged) in the chamber, (c) gas-phase (solid lines, normalized to reagent ions at first and then to their maximums) and particle-phase (dashed lines, normalized to reagent ions) HOM species (10-s averaged), (d) total aerosol, organics, and sodium chloride mass concentrations from SMPS (2.2-min averaged, black circles) and AMS measurements (20 s averaged). The first and second shaded areas are gas phase and particle phase background measurements, respectively. Note that the time series of NaCl was estimated using the method explained in the supplementary owing to the lack of measurements, and no measurements given above were corrected for chamber wall loss."*

**Comment #7:**

Line 192: I think that the logic of this sentence is somewhat odd. Or did you mean more volatile components decrease already at the "lowest" temperatures?

**Response:**

Yes, it may be the case that more volatile components decrease already at the "lowest" temperatures as the reviewer suspected. Based on our previous results by Häkkinen et al. (2022), we found that the sum of HOM monomers peaks at 120 ℃.

Here, we were trying to convey that the temperature that was chosen for the measurements could largely affect the distribution of observed HOM compounds. Although we focused on the results obtained at

230 ºC, which detects the largest fraction of SOA mass, higher temperatures may falsely "concentrate" dimers because 230 ºC is not optimal for monomer measurements (lower by a factor of ~1 vs. 120 ºC for monomers). This bias needs to be kept in mind when interpreting the particle-phase HOM mass spectra. In order to make this point clear, we rewrite this sentence as follows:

*"However, as shown in Fig. 2c, larger molecules ($C_{17}$ and $C_{20}$ compounds) were found to evaporate more efficiently at 230 ºC, while signals of more volatile molecules (e.g. $C_{10}$ compounds) were already decreasing at this temperature. This effect, where the choice of evaporation temperature significantly impacts the distribution of observed species, is consistent with the previous results reported by Häkkinen et al. (2022) and needs to be kept in mind when interpreting the particle-phase HOM mass spectra."*

**Comment #8:**

Line 214: I don't understand what you want to say here.

**Response:**

By showing that the nitrate dimer ($HNO_3 \cdot NO_3^-$) charged signals contributed only a small fraction to the total signal ($4.3 \pm 1.5\%$), it would be safe 1) if we only focus on the $HOM_{CHO}$ species in the experiments without $NO_x$ addition, and 2) to assume that the N-containing $HOM_{ON}$ species measured in the $NO_x$ experiments were mainly formed through $NO/NO_3$ reactions instead of nitrate dimer changed $HOM_{CHO}$ species. To make this point clear, we modified this sentence:

*"Note that a relatively low contribution of nitrate dimer ($HNO_3 \cdot NO_3^-$) charged signals to the total signal ($4.3 \pm 1.5\%$) was observed for α-pinene ozonolysis experiments. Thus, we only focused on $HOM_{CHO}$ compounds in the experiments without $NO_x$ addition and the $HOM_{ON}$ species measured in the $NO_x$ experiments were mainly formed through $NO/NO_3$ reactions instead of nitrate dimer charged $HOM_{CHO}$ compounds".*

**Comment #9:**

Line 450: There is probably not much NO left in the $NO_3$ case, therefore no termination with NO. "All" $NO_X$ should be $NO_2$, or?

**Response:**

Yes, the reviewer is correct. In the NO case, photolysis of $NO_2$ was used to introduce NO into the chamber. On the other hand, in the $NO_3$ case UV lights are off during the entire experiment, thus all $NO_x$ should be $NO_2$.

**Comment #10:**

Line 454: "Here", and "in this work" is redundant.

**Response:**

We deleted "Here" and left "in this work" in the manuscript.

**Comment #11:**

**Response:**

Yes, it is the time series of gas-phase HOM monomers and dimers as well as the dimer/monomer ratio during an experiment. Both the HOM monomers and dimers decreased after the addition of NaCl seed particles, but the relative change of the dimers was larger, and therefore the dimer/monomer ratio also decreased. The right y-axis may have been a little confusing, so we changed it to a linear scale.

[Figure]

*"Figure S7. (d) Time series of gas-phase HOM monomers and dimers as well as the dimer/monomer ratio measured in experiment No. 1 (input flow with 53 ppb α-pinene and 33 ppb $O_3$)."*

**Comment #12:**

**Response:**

We replaced "detect" with "detected".

**Comment #13:**

**Response:**

We added "are" in front of "compared".

**Comment #14:**

**Response:**

We changed "$C_{10}H_{15}O_8$" to "$C_{10}H_{15}O_{8-9}$".

**Comment #15:**

Line 621: much "shorter" ?

**Response:**

We replaced "short" with "shorter".

**Comment #16:**

Figure S8: On which axis read the pink triangles?

**Response:**

The pink and purple markers share the right axis. We changed this figure to make it clear.

[Figure]

*"Figure S8. Ratio of D/M ratios obtained between different phases (using the same dataset as used in Figure. 8c)."*

**Comment #17:**

Table S2 and S3: The readability of these tables could be possibly improved by comparing only the same CxHy in one line, allowing for gaps where other methods did not find this class of compounds.

**Response:**

We changed Table S2 and Table S3 as below according to the reviewer's suggestion.

*"Table S2. Comparison of main particle-phase compounds (by α-pinene and O₃ reactions) identified by a NO₃-CIMS used in this work and other techniques. The chemical formula of main monomers and dimers are roughly listed according to their relative abundance in SOA (e.g. O numbers), if the concentration was measured/reported."*

[revised manuscript text omitted]

**Comment #18:**

Table S3 header: listed

**Response:**

We replaced "list" with "listed".

**Comment #19:**

Table S4: Not easy to read. It could help if you convert the references to symbols, which you list under the table and remove the line breaks in the columns with number ranges.

**Response:**

Thanks for the suggestion. We made some modification on Table S4 and made Figure S10 to clearly show the comparison between different studies.

*"Table S4. Summary of elemental ratios of SOA and HOM (formed from ozonolysis of α-pinene under various conditions) measured in both the gas and particle phases."*

| description | gas phase CI-APi-TOF | | particle phase AMS | | | | references |
|---|---|---|---|---|---|---|---|
| | H/C | O/C | H/C | O/C | SOA | method | |
| AP + $O_3$ | / | / | 1.38-1.51 | 0.29-0.46 | 0.5-140 | AA | (Shilling et al., 2009) |
| | / | / | 1.47 | 0.43 | 57-183 | AA | (Chhabra et al., 2010) |
| | 1.5 | 0.7 | 1.5 | 0.6 | 5-10 | IA | (Ehn et al., 2014) |
| | 1.5 | 0.8 | 1.4-1.6 | 0.44-0.76 | / | IA | (Claflin et al., 2018) |
| | / | / | 1.59-1.71 | 0.26-0.56 | 6-100 | IA | (Jensen et al., 2021) |
| | / | 0.7-0.81 | / | / | / | / | (Molteni et al., 2019) |
| | 1.51 | 0.79-0.8 | 1.48-1.53 | 0.43-0.45 | 3-15 | IA | This study |
| | | | | | | | |
| AP + $O_3$ + CO | 1.51 | 0.79 | 1.53 | 0.43 | ~10 | IA | This study |
| AP + $O_3$ + $NO_x$ | 1.51 | 0.78-0.83 | 1.52-1.54 | 0.34-0.42 | 3-10 | IA | This study (lights off) |
| AP + $O_3$ + $NO_x$ | 1.48 | 0.84-1.02 | 1.53-1.54 | 0.40-0.48 | 2-15 | IA | This study (lights on) |
| | | | | | | | |

*"Notes: elemental ratios adopted from previous studies using the AA method (Aiken et al., 2008) were converted to IA methods by scaling an empirical factor of 1.27 and 1.11 for O/C and H/C ratios, respectively (Canagaratna et al., 2015), in Figure S10. SOA mass in above table was given in unit of ug m$^{-3}$."*

[Figure]

*"Figure S10. Summary of elemental ratios of SOA and HOM (formed from ozonolysis of α-pinene under various conditions) measured in both the gas and particle phases. The detailed description is given in Table S4."*

---

## Author Comment (AC2)

**Response to Reviewer #2**

**General comment**

Understanding the molecular level of aerosol formation has been a long-standing challenge, especially concerning the detailed chemical composition of organic aerosols. Zhao et al. used a new VIA-nitrate-CIMS technique to measure the detailed chemical composition of SOA produced by ozonolysis of α-pinene and provided some interesting new insights. They found the detected HOM molecules in the aerosol phase are significatnly different from the "condensed" phase, especially in the dimer range, and indicated that the aerosol phase reactions influenced the aging processes. The manuscript is overall well organized. And more importantly, this is an encouraging attempt and can be an important step in understanding organic molecules in the aerosol phase. I recommend it can be published after a minor revision.

We thank the reviewer for the positive and insightful comments, and we answer the specific comments point-by-point below. The reviewer's comments are in **blue**, and our answers are in **black**.

**Specific comments**

**Comment #1:**

The main findings from this work are, to some extent, similar of Pospisilova et al., 2020. Both studies suggest that aerosol phase reaction plays important role in the aging processes, and the decay of $C_{20}$ and $C_{19}$ HOM dimers was likely the source of $C_{17}$ dimers in the aerosol phase. I suggest that the authors can discuss more on the difference from Pospisilova et al., 2020.

**Response:**

The gas- and particle-phase HOM measurements in Pospisilova et al., (2020) were conducted with different ionization methods. In this work, we used a nitrate chemical ionization inlet for both the gas and particle phase HOM measurements to minimize the uncertainty during comparisons. In addition, we introduced CO or $NO_x$ into the α-pinene ozonolysis reaction system to change the distribution of gas-phase HOM products and investigated the impacts on both phases.

Based on the suggestion of the reviewer, we have now modified/extended the discussion concerning the comparison of our VIA system to the EESI system to emphasize the difference between this work and Pospisilova et al., 2020, in line 357: *"Furthermore, particle phase decomposition of more carbon-containing HOM compounds (e.g. $C_{10}$ in the monomer range and $C_{20}$ in the dimer range) was proposed as an important decay pathway of α-pinene SOA (Pospisilova et al., 2020), which could largely explain the relative enhancement of $C_{16}$-$C_{19}$ compared to $C_{20}$ compounds. However, the lifetime of SOA particles is different between the two studies. For example, $C_{16}$-$C_{17}$ signals peak after ~2 hours of the α-pinene ozonolysis in their batch-mode experiments, whereas the residence time is ~1 hour for the continuous-mode experiments conducted in this work. Nevertheless, about half of the compounds were proposed to have half-lives of less than one hour (Pospisilova et al., 2020). If the thermal energy supplied during the evaporation of SOA within the VIA can promote this decay process, then the observed particle-phase HOM distribution would be expected to shift to lower C numbers."*

And we modified the sentence in line 380: *"In order to understand this "dimer enhancement" better and to extend the results from α-pinene ozonolysis system (Pospisilova et al., 2020) to various conditions, we conducted experiments where we tried to modify the dimer/monomer (D/M) ratios of HOM in the gas phase, by adding CO or NO to the system (as discussed in the following sections), to investigate the effects of the corresponding D/M ratios in the particles."*

My biggest concern is still how much we can trust the VIA measurement. Is it possible the heating in the inlet speed up the decay of condensed HOM dimers? The authors suggested some possible reactions that should be responsible for the particle phase processes, e.g., the Baeyer-Villiger reactions. Does the heating process have the potential to influence the Baeyer-Villiger reactions?

**Response:**

The largest uncertainty of the current VIA inlet is indeed the inherent limitations of the thermal desorption techniques itself, i.e thermal decomposition, especially when we care about the chemical composition of the evaporated vapors. Another method would be extraction if one wants to avoid thermal decomposition. However, the solvent used (e.g. water or some organic compound) may be selective towards some specific functionalities and may introduce a matrix effect. That is partly why the thermal evaporation method is still one of the widely used techniques since no perfect technique exists for now.

Nevertheless, we found that with the much shorter residence time (~0.2 s) in the VIA desorption tubing, less thermal decomposition seems to take place (which looks like very promising) compared to, e.g. the FIGAERO measurements (Lopez-Hilfiker et al., 2014; Lopez-Hilfiker et al., 2015). If thermal decomposition did play an important role in this VIA-$NO_3$-CIMS system, we would expect more decomposition of dimers (labile RO-OR compounds) than monomers, and in particular at the highest temperature. However, we observed the opposite, i.e. high dimer/monomer ratios in the particle phase. Although the measurements at 230 ºC may falsely "increase" dimers but "lose" monomers compared to the measurements at 170 ºC (Figure 2), it is still far from enough to explain the enhancement of dimers in the particle phase (Figure 8b). Concerning the opposite effect, as the reviewer might have been suggesting, that the heating would enhance accretion product formation, we would expect that this effect would be much stronger in systems like the FIGAERO where the particles spend much longer periods at the elevated temperatures. The short heating period in the VIA should minimize the probability of such reactions.

It remains an open question if (and to what extent) the thermal decomposition of SOA happens under different temperatures and if the thermal desorption techniques could promote particle-phase reactions. In order to understand more on this topic, we are making a more detailed characterization of the VIA-NO3-CIMS system, trying to understand the temperature-dependent sensitivity (including evaporation and transmission efficiencies) for different types of organic molecules. This will at least narrow down part of the above uncertainties the reviewer asked about.

**Comment #3:**

Please provide the NO concentration for all the NOx runs

**Response:**

In the NO runs, photolysis of $NO_2$ was used to introduce NO into the chamber. Thus, only the initial concentrations of $NO_2$ were provided in Table 1. After the reaction system reached a steady state, the NO concentrations ($[NO]_{ss}$) were measured and summarized in Table 1.

**Comment #4:**

*Although I understand this is no longer possible in this work, it would be interesting to compare the three technologies, EESI, VIA, and FIGAERO in an experiment together.*

**Response:**

We fully agree that it would be very interesting to have the currently widely used FIGAERO and EESI systems in parallel with the VIA-$NO_3$-CIMS for experiments, testing e.g. various reaction conditions. But unfortunately, we did not have an EESI or a FIGAERO available during the experiments. We hope that we can contribute to such an intercomparison in the future.

**Reference:**

Pospisilova, V., Lopez-Hilfiker, F. D., Bell, D. M., El Haddad, I., Mohr, C., Huang, W., Heikkinen, L., Xiao, M., Dommen, J., Prevot, A. S. H., Baltensperger, U., and Slowik, J. G.: On the fate of oxygenated organic molecules in atmospheric aerosol particles, Science Advances, 6, eaax8922, doi:10.1126/sciadv.aax8922, 2020.

Lopez-Hilfiker, F. D., Mohr, C., Ehn, M., Rubach, F., Kleist, E., Wildt, J., Mentel, T. F., Lutz, A., Hallquist, M., Worsnop, D., and Thornton, J. A.: A novel method for online analysis of gas and particle composition: description and evaluation of a Filter Inlet for Gases and AEROsols (FIGAERO), Atmos. Meas. Tech., 7, 983-1001, 10.5194/amt-7-983-2014, 2014.

Lopez-Hilfiker, F. D., Mohr, C., Ehn, M., Rubach, F., Kleist, E., Wildt, J., Mentel, T. F., Carrasquillo, A. J., Daumit, K. E., Hunter, J. F., Kroll, J. H., Worsnop, D. R., and Thornton, J. A.: Phase partitioning and volatility of secondary organic aerosol components formed from α-pinene ozonolysis and OH oxidation: the importance of accretion products and other low volatility compounds, Atmos. Chem. Phys., 15, 7765-7776, 10.5194/acp-15-7765-2015, 2015.